# Observational evidence of increased afternoon rainfall downwind of irrigated areas

P. Greve [1] ✉, A. U. Schmitt[1], D. G. Miralles [2], S. McDermid [3,4], K. L. Findell [5], A. García-García [6] & J. Peng [6,7]

Irrigation plays a vital role in addressing the growing food demand of an increasing global population. It represents one of the most critical and direct human interventions on the coupled water and energy cycles. As irrigated farmland continues to expand, understanding the climate impact of extensive irrigation becomes increasingly important. Yet, the effect on rainfall patterns near irrigated areas remains unclear. Here, using two global, high-resolution, sub-daily precipitation datasets, we show that afternoon rain events occur more often 10 km to 50 km downwind and less often upwind of extensively irrigated land. However, we also find that the total amount of heavy afternoon rain downwind of irrigated areas is lower than upwind. Our results establish large-scale observational evidence of the local precipitation dynamics surrounding irrigated areas; these insights will help constrain the representation of these processes in next-generation climate and weather forecasting models and provide valuable insights for regional water management.

About 70% of worldwide freshwater withdrawals are used for irrigation, and of the ca. 16 million km² of global cropland, about 20% are irrigated[1]. Irrigation is essential to sustain food production in some of the most populated regions of the world. However, it can also exacerbate water scarcity and deplete groundwater resources[2–4], leading to serious consequences for groundwater and river chemistry[5]. Due to the massive redistribution of water across the land surface and pumping of groundwater resources, irrigation further represents a significant human intervention in the water and energy cycles. Essentially, irrigation increases soil moisture in irrigated fields, affecting the water and energy balance by enhancing evaporation, reducing surface temperature and land surface albedo, and thus affecting the radiative balance and sensible and ground heat fluxes. As a result, large-scale, extensive irrigation alters regional land–atmosphere interactions. Both observations and modelling-based assessments indicate that a decreased sensible heat flux due to irrigation cools daytime surface air

temperatures[6–10], particularly under heatwave conditions[11–13]. However, enhanced atmospheric moisture due to irrigation can intensify moist heat stress[14] and increase nighttime temperatures[15,16].

Local field campaigns, like GRAINEX[10,17–20] in the midwestern United States and LIASE[21,22] in northern Spain, have specifically investigated the potential impact of irrigation activities on local atmospheric (thermo-)dynamics. In both campaigns, cooler daytime temperatures and lower atmospheric boundary layer heights were found under irrigated conditions, stabilising the atmosphere[10,21]. However, the resulting impact on local and regional precipitation remained less clear. Although these studies suggest that extensive irrigation increases atmospheric stability over irrigated areas, enhanced evaporation also creates moisture gradients that may alter regional moisture convergence[18,22]. Increased atmospheric water vapour may additionally be embedded within the large-scale air flows, leading to moisture convergence away from the irrigated areas. Model

[1]Climate Service Center Germany (GERICS), Helmholtz-Zentrum Hereon, Hamburg, Germany. [2]Hydro-Climate Extremes Lab (H–CEL), Ghent University, Ghent, Belgium. [3]Department of Environmental Studies, New York University, New York, NY, USA. [4]NASA Goddard Institute for Space Studies, New York, NY, USA. [5]Geophysical Fluid Dynamics Laboratory (GFDL), National Oceanic and Atmospheric Administration (NOAA), Princeton, NJ, USA. [6]Department of Remote Sensing, Helmholtz Centre for Environmental Research–UFZ, Leipzig, Germany. [7]Institute for Earth System Science and Remote Sensing, Leipzig University, Leipzig, Germany. ✉e-mail: peter.greve@hereon.de

results based on the increasing number of (regional) climate models representing irrigation activities[7,8,11–13,23–28] remain highly uncertain and suggest a variety of mesoscale impacts on both seasonal precipitation and convective rain events. For example, across the Central Valley in California, vapour transport from irrigated areas towards the Sierra Nevada is triggered, resulting in a general precipitation increase[29]. A potential explanation is that daytime cooling increases the thermal contrast between mountains and adjacent irrigated plains, influencing local wind patterns. This mechanism could also explain an increase in afternoon precipitation in mountain ranges close to the North China Plain[30]. Modelling experiments further indicate an increase in mean monthly rainfall downwind of irrigated areas within the Sahel region[25]. However, using a regional climate model, upwind increases in rainfall were detected in Western Africa[31]. Considering the event-scale, convection-permitting regional climate simulations of the 2015 summer heatwave in Northern Italy indicate that afternoon rain events were inhibited under irrigated conditions, also in the nearby Alpine regions[32]. Analysing a single rain event during the GRAINEX field campaign using a regional climate model showed that precipitation intensity generally decreased under irrigated conditions[19]. Regional climate models further indicate that irrigation leads to an anomalous pressure field surrounding heavily irrigated areas in Saudi Arabia, resulting in local precipitation decreases over irrigated areas and enhanced precipitation away and possibly upwind from the irrigated regions[33]. Additionally, simulated evidence exists for synoptic-scale and remote impacts of irrigation on monsoon dynamics[24,26] and regional circulation patterns[34].

The wide range of results obtained from climate models could be associated with model uncertainties, highlighting the need to constrain the simulations with observations. Increases in mean monthly rainfall downwind of irrigated areas within the Sahel region can be confirmed by observations, even though underlying uncertainties remain substantial[25]. Observational evidence across the Midwestern United States further suggests an increase in summertime precipitation intensity and total precipitation downwind of extensively irrigated areas. Rain intensities specifically increased between 1950 and 1980, coinciding with a rapid expansion of irrigation activities in the region[35].

The impact of surface moisture anomalies (like irrigation) on convective afternoon rain events is mediated through daytime boundary layer processes. Spatial gradients in soil moisture have been shown to lead to the development of mesoscale circulations with ascending motion and increased afternoon rainfall over drier areas where sensible heat fluxes are enhanced relative to adjacent wetter areas[36,37]. On the other hand, considering temporal variability in surface fluxes shows that days with high evaporation have an increased likelihood of convective afternoon rain in the eastern US[38], consistent with the observed development of favourable convective initiation characteristics across irrigated areas in the midwestern United States during the GRAINEX field campaign[20]. These findings align when considering both the temporal and spatial variability of soil moisture, as afternoon rainfall is most likely over the driest patches of otherwise relatively wet and heterogeneous soil moisture conditions[39]. However, the preference for an increased likelihood of afternoon rain over drier or wetter soils is further complicated by the stability and humidity of the near-surface air entrained by the growing daytime boundary layer[40], as well as by atmospheric dynamics and moisture convergence[41].

Despite the variety of regional studies focused on investigating the impacts of irrigation, large-scale, observation-based studies are still lacking, limiting the assessment of the impact of irrigation on precipitation patterns at a global scale. However, the increasing wealth and quality of satellite Earth observations currently enables the mapping of irrigation crop areas and the identification of precipitation episodes on a global scale. As such, a global analysis of the effect of irrigation on regional precipitation patterns becomes possible.

Here, we investigate the impact of large-scale irrigation on the likelihood of afternoon rain on and around irrigated land worldwide. We use two global, high-resolution (0.1°), sub-daily precipitation datasets and reanalysis-based wind direction to determine the location of individual afternoon rain events relative to nearby irrigated land. The precipitation data are (i) multi-source weighted-ensemble precipitation (MSWEP) 3-hourly precipitation[42,43] and (ii) the integrated multi-satellitE retrievals for GPM (IMERG) half-hourly precipitation dataset[44]. We estimate afternoon rain as the total amount of rain between 12h and 24-h local time. By focusing on localised afternoon rain events and using temperature thresholds as proxies for growing seasons, we aim to understand how irrigation may influence afternoon rainfall within a 50-km radius surrounding irrigated areas. Please refer to Fig. 1 and the Methods for detailed information on the selection criteria, thresholds and data sources.

Considering the global distribution of irrigated areas[45], most afternoon rain events near irrigated grid cells are detected in regions with extensive paddy irrigation, such as South Asia (see Fig. 2). However, about 52% (MSWEP) and 61% (IMERG) of the detected rain events occur outside South Asia, primarily in East and Southeast Asia (e.g. Thailand, Cambodia, Vietnam), North America, Southern and Eastern Europe and Australia. It is important to note that the following analysis excludes irrigated coastal and mountainous areas, such as many regions in China, Central Asia, Southern Europe and California, given the selection criteria to omit afternoon rain events dominated by orographic effects or nearby oceans.

## Results

### Afternoon rain and irrigation: global signals

For each rain event, we first locate the grid cell with the peak afternoon rainfall; we then determine the relative position of the irrigated grid cells within a wind-based reference system. Due to the lack of global, satellite-based wind data over land surfaces within the study period, we use ERA5 10-m wind data[46] to compute a monthly wind direction climatology from 2001 to 2020. This approach enables us to assess whether afternoon rain is more likely to occur upwind or downwind of irrigated grid cells. We refrain from using daily or sub-daily wind directions obtained from ERA5 because the land surface component of ERA5 does not consider irrigation. This omission could result in erroneous wind estimates when explicitly looking at grid cells subject to irrigation in real life. By using monthly climatological winds, we focus on large-scale synoptic moisture transport rather than sub-daily mesoscale circulations influenced by local land–atmosphere interactions. More detailed information on the wind-based reference system and the likelihood assessment is provided in the Methods.

We find that the location of maximum afternoon rain is more likely to be located downwind of irrigated grid cells (see Fig. 3). Especially when considering IMERG rain data, peak afternoon rain is less likely to be located upwind of the irrigated grid cells. This further indicates a likelihood of peak afternoon rain directly at the location of the irrigated grid cells which falls between the likelihood in downwind and upwind areas. Using randomly sampled wind directions (see Methods, dashed line and pale shading in Fig. 3c, d) as a benchmark enables us to better assess the impact of actual wind direction on afternoon rain likelihood. Assuming random wind directions, the probability of maximum afternoon rain using IMERG decreases with increasing distance from the irrigated grid cell. This means that the smaller positive downwind signals identified using actual wind direction data in IMERG, compared to the larger downwind signal using MSWEP, become more pronounced. This is highlighted in a normalised version of Fig. 3c, d provided in Supplementary Fig. S3. In addition, while the overall signals are small (<10%), it is evident that they clearly emerge from the confidence interval we obtained from randomly sampling wind directions. This suggests a robust impact of irrigation

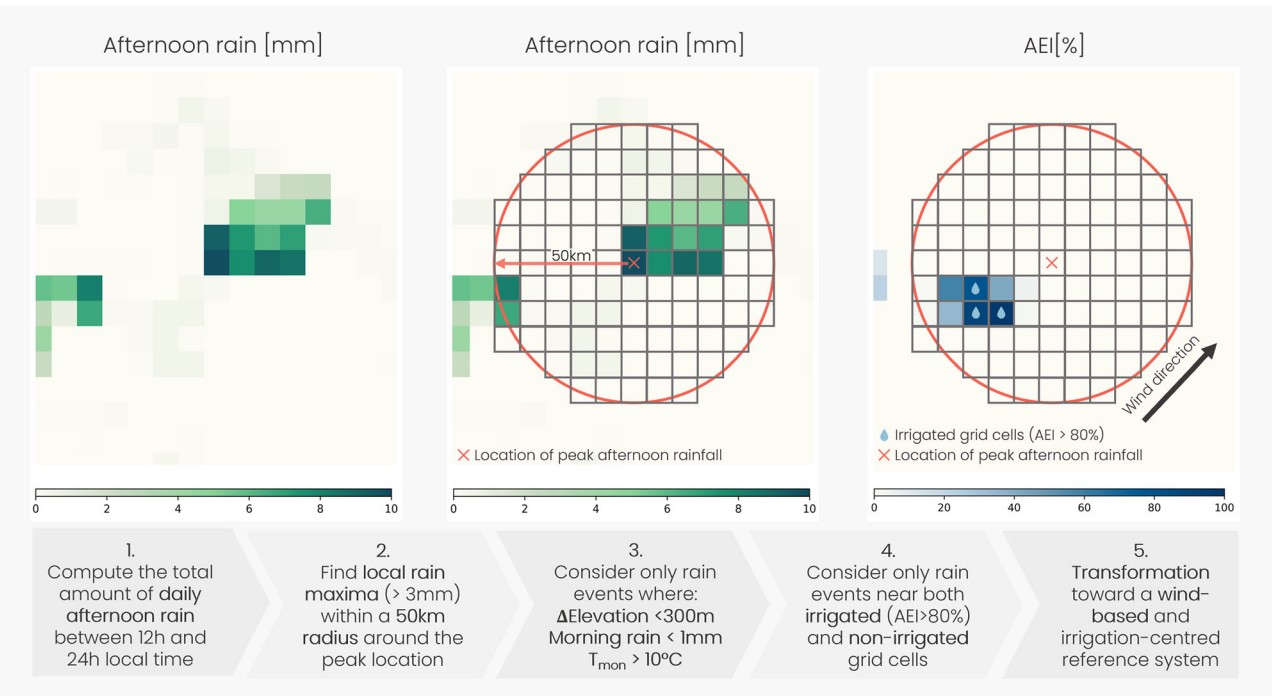

**Fig. 1 | Afternoon rain event detection.** Illustration depicting the detection of an afternoon rain event featuring irrigated grid cells (AEI area equipped for irrigation[45]) within a 50-km radius surrounding the peak location. Please refer to the Methods for more information on Step 5. $T_{mon}$ represents the average monthly temperature.

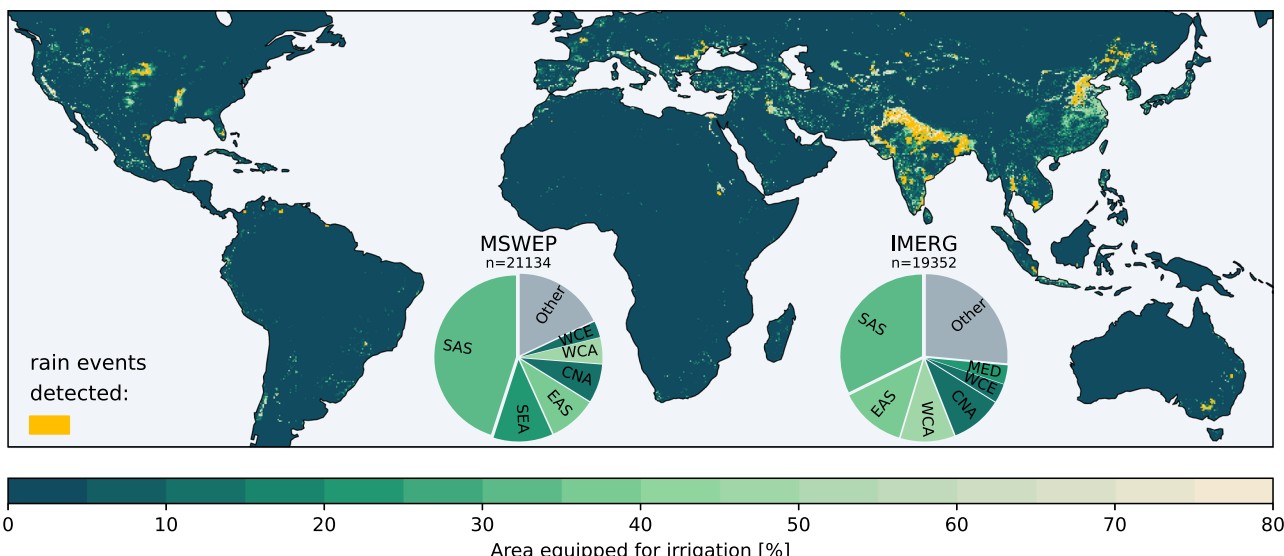

**Fig. 2 | Global map of irrigated areas and detected afternoon rain events.** Global map showing the percentage of area equipped for irrigation (AEI) within each grid cell. Highlighted regions indicate areas where afternoon rain events (see Fig. 1 and Methods) occur over and close (<50 km) to irrigated grid cells with AEI >80% based on either MSWEP or IMERG rain data. Pie charts depict the distribution of these rain events across IPCC regions (see Methods and Fig. S1) Colours of individual sectors indicate the percentage of irrigated cropland relative to the total cropland area within the respective region. Please refer to Fig. S2 for a map showing AEI without overlapping rain event locations.

and prevailing winds on the likelihood of afternoon rain surrounding irrigated areas at the global scale.

### Afternoon rain and irrigation: regional signals

The majority of considered rain events are located in South Asia (MSWEP - ca. 48%, IMERG - ca. 39%, see Fig. 2 and Fig. S1). To assess whether conditions in South Asia dominate the overall signal, we divided the sample into rain events in South Asia and outside (see Fig. 4a, b). The comparison shows that similar signals are found within and outside South Asia, suggesting that the obtained global signal is not predominantly influenced by mechanisms shaping irrigation–rainfall interactions in South Asia. The decreased likelihood of peak afternoon rain upwind of irrigated grid cells is even more pronounced outside South Asia, especially when considering MSWEP data. Furthermore, the MSWEP data reveal a distinct maximum in the likelihood of peak afternoon rain ~2 to 3 grid cells (ca. 20 to 30 km) downwind of irrigated grid cells is present outside South Asia. In individual regions outside South Asia, the sample sizes are

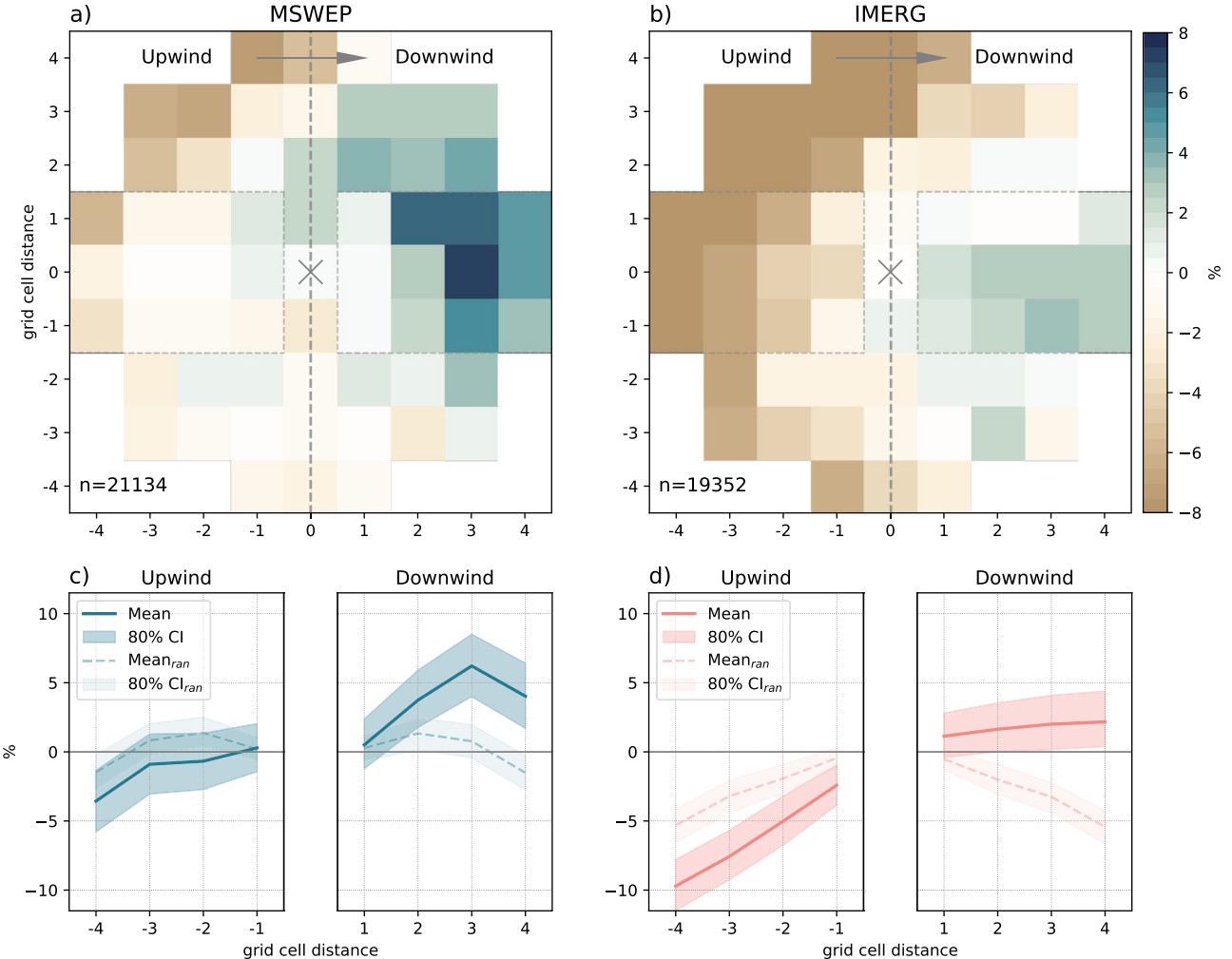

**Fig. 3 | Location of peak afternoon rainfall relative to irrigated grid cells.**
**a**, **b** Raster maps showing the likelihood of the peak location of afternoon rain relative to irrigated grid cells centred at (0,0) for MSWEP and IMERG rain data. Wind direction is normalised such that upwind (downwind) areas are on the left (right). Likelihood is computed by comparing the number of peak rain events at each position to the average number of peak rain events at (0,0) (see Methods). Blue (brown) colours denote an increased (decreased) likelihood of peak rain events relative to the location of irrigated grid cells. **c**, **d** Average likelihood of peak afternoon rain along the wind direction (±1 grid cells upwind and downwind bands, within the dashed horizontal lines in **a**, **b**). The shaded area represents the 80% confidence interval (CI) based on a bootstrapping approach (see Methods). The dashed line and the pale shading indicate the average and the 80% CI for the randomised wind directions (CI_ran).

much smaller, leading to significant uncertainties (Fig. S4). While regional responses across North America, South/Central Europe and East Asia are consistent with the global signal, results in other regions remain inconclusive. These areas require more detailed assessments, explicitly considering local and regional meteorological, climatic and non-climatic conditions.

We further divide the full sample, including all rain events, into two equally sized samples of rain events that occur under either more arid or humid conditions (see Methods). Our results show that the increased likelihood of afternoon rain downwind (MSWEP) and decreased likelihood upwind (IMERG) of irrigated grid cells is most pronounced under more arid conditions (see Fig. 4c, d). Over these regions, irrigation is applied more frequently, and the contrast in land surface properties and fluxes to surrounding dry areas is more pronounced. While no significant signal is detected under more humid conditions, a weak signal may still be detected compared to randomly sampled wind directions.

**Irrigation impact on total afternoon rain amount**
We further explore whether the increased likelihood of rain events downwind of irrigated grid cells affects rainfall amounts (in mm/12h,

i.e. the total amount of rain between 12 and 24 h at the location of the afternoon rain maximum). We compare probability distributions using quantile-quantile plots (Q-Q plot, see Fig. 5) based on the comparison of two samples: (i) all rain events located in the downwind band of irrigated grid cells (see Fig. 3) and (ii) all non-downwind rain events located in other areas surrounding irrigated grid cells. Compared to afternoon rain events downwind of irrigated grid cells using MSWEP data, non-downwind rain events show slightly higher rain amounts for some of the strongest events (above Q90). A much more pronounced signal can be detected using IMERG rain data, showing significantly higher rain amounts above Q50 for non-downwind rain events. For both MSWEP and IMERG, the Q-Q plot is slightly steeper than the 1:1 line, indicating that the non-downwind sample distribution could be more dispersed. We assess uncertainties using a Monte Carlo approach (see Methods). While the magnitude of differences between MSWEP and IMERG above Q90 differs, both Q-Q plots more clearly emerge from the envelope of uncertainty for the highest quantiles. Our findings, therefore, suggest a distinct local impact on the total amount of afternoon rain, indicating lower downwind rainfall amounts during the most extreme events compared to those in other areas surrounding irrigated grid cells.

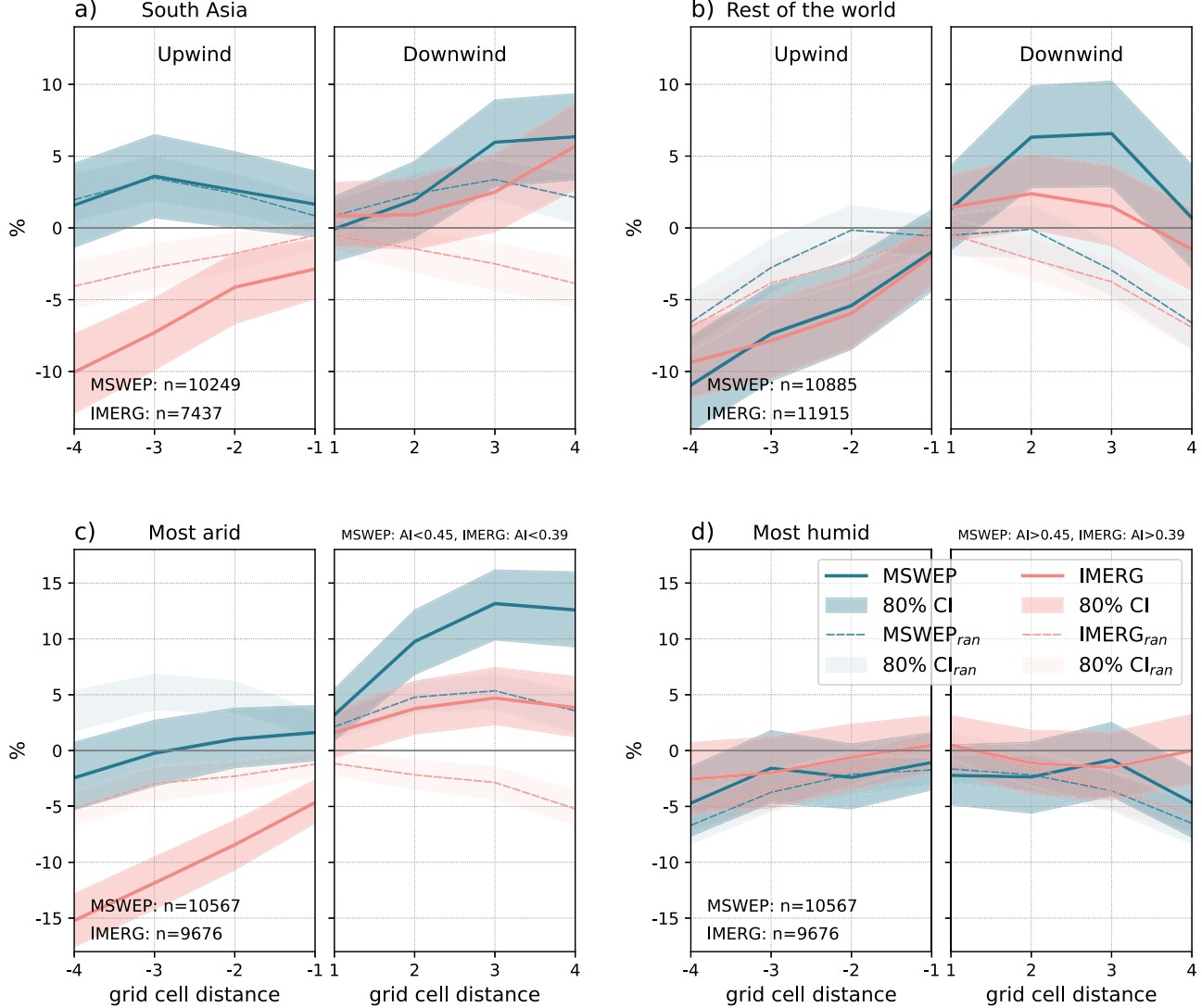

**Fig. 4 | Regional differences in peak rainfall likelihood.** Average likelihood of peak afternoon rain along the wind direction (same as in Fig. 3) for rain events detected in **a** South Asia only, **b** outside of South Asia, **c** the more arid and **d** the more humid half of all rain events split based on the provided aridity index (AI) values. The shaded area represents the 80% confidence interval (CI) based on a bootstrapping approach. The dashed lines indicate the 80% CI for the randomised wind directions (CI_ran).

## Exploring mechanisms in irrigation–rainfall interactions

This work shows that afternoon rain is more likely to occur downwind of irrigated grid cells globally, suggesting an influence of irrigation activities on regional precipitation patterns. Similar precipitation–irrigation interactions have been observed downwind of irrigated areas in regions such as the eastern Sahel[25] and the Midwestern United States[35]. However, modelling experiments indicate a more varied response of local precipitation patterns to irrigation. These responses range from upwind[31,33] to downwind rainfall increases[25,29], and from less intense[19,30,32,33] to enhanced local rainfall directly over irrigated areas[23].

Our results suggest that additional moisture added to the atmosphere through enhanced evaporation under irrigated conditions is more likely to affect downwind rain rather than causing rain directly over irrigated areas. This hints toward an increased convective inhibition and an overall stabilisation of the atmosphere directly over irrigated areas due to reduced surface air temperatures. Recent field campaigns determined a similar mechanism primarily related to shallower and more stable boundary layers under irrigated conditions[10,21]. Afternoon rain is still more likely over irrigated grid cells than upwind,

indicating that increasing atmospheric moisture through enhanced evaporation has at least some local effect, which partly corresponds to observations of convective properties across irrigated areas within the midwestern United States[20]. In addition, the detected upwind gradient (especially for IMERG) indicates that afternoon rain is becoming more likely closer to irrigated areas. One possible explanation is that irrigation alters mesoscale wind patterns, leading to moisture convergence upwind of irrigated areas[31,33]. However, most detected rain events include at least a few additional irrigated grid cells within the 50-km circle surrounding the peak location (see Fig. S5). As a result, many rain events feature irrigated grid cells both upwind and downwind of their peak location. Therefore, while our analysis identifies a strong (positive) downwind signal, we must also anticipate a corresponding (negative) upwind signal to some extent.

However, our results primarily suggest that irrigation-induced moisture advection causes an increased likelihood of afternoon rain downwind of irrigated areas. While morning boundary layer growth over irrigated areas is likely dampened due to reduced surface heating, boundary layers can grow unhindered in dry nearby non-irrigated regions as most available energy is partitioned toward sensible heat[10].

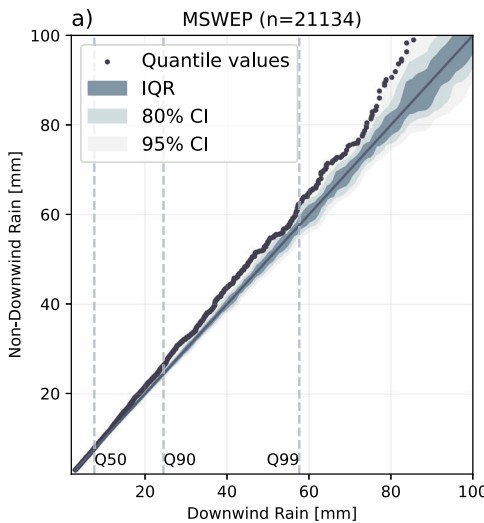

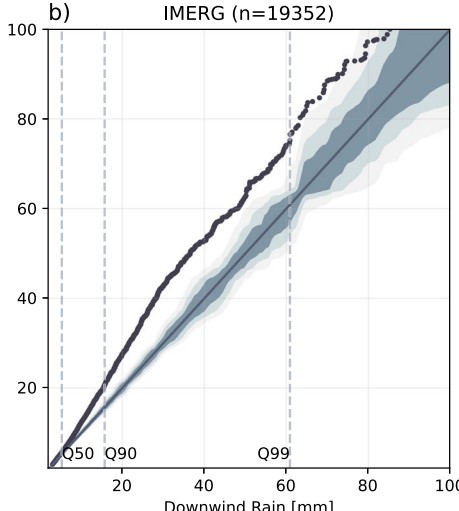

**Fig. 5 | Downwind vs. non-downwind rainfall amounts.** Quantile-quantile plots of the total amount of afternoon rain per event downwind (x-axis) of irrigated grid cells (downwind band in Fig. 3) and in the non-downwind areas surrounding irrigated grid cells (y-axis) considering **a** MSWEP and **b** IMERG rain data. Individual grey dots represent rainfall events. The shadings represent the interquartile range (IQR), the 80% and the 95% confidence interval (CI). Uncertainties are estimated by choosing 1000 equally sized random samples. The vertical dashed lines illustrate the 50th, 90th and 99th quantile (Q50, Q90 and Q99). The diagonal black line represents the 1:1 line indicating a perfect match of the sample distributions.

Entraining additional moisture into these conditions due to moist air advection from irrigated areas during the day may increase vertical mixing in the convective boundary layer. Moist air is less dense, which enhances the efficiency of convective thermals, leading to increased upward motion and potentially triggering deep convection. Possible nighttime advection of moist air from irrigated areas may be weakened[18], but can still increase morning atmospheric moisture levels in downwind non-irrigated areas. This can result in an earlier onset and more rapid boundary layer growth—additionally enhancing the potential for deep convection[47]. These processes are particularly pronounced under dry conditions in the surrounding non-irrigated areas, leading to an even stronger surface heating. In addition, higher evaporative demand under arid conditions further enhances evaporation and transpiration across irrigated areas. Our detected signal of higher downwind afternoon rain likelihood is most pronounced across the most arid regions worldwide (see Fig. 4) and is consistent with the described boundary layer dynamics. These results align with previous studies suggesting a higher likelihood of afternoon rainfall over drier soils adjacent to wetter soils[36,39] and a higher likelihood of afternoon rain with higher evaporation rates[38]—suggesting that these findings also apply to irrigated areas globally.

Our findings indicate that while afternoon rain downwind of irrigated areas is more likely, the total rainfall amounts are lower than those in the non-downwind areas surrounding irrigated grid cells, especially for the most extreme events (above Q90, see Fig. 5). One potential explanation for our findings is that moisture in combination with lower temperatures from upwind irrigated areas is more likely to initiate shallower convection, resulting in more frequent but less intense rain. At the same time, irrigation may lead to daytime mesoscale circulations, which can cause convergence upwind of irrigated areas[22]. Additionally, if convection is triggered in areas outside the cooler and moister downwind regions, it is more likely to happen under highly unstable atmospheric conditions, promoting deeper convection. These intense storms can create inflow patterns that draw in additional moisture from nearby irrigated areas, enhancing storm intensity.

It is important to note that a better understanding of these processes requires high-resolution and more detailed data on near-surface wind patterns, atmospheric moisture, boundary layer characteristics and land surface information. Many of these data are currently not available at the scales of this study. However, we encourage upcoming research to investigate these processes in more detail in order to gain much-needed insights into irrigation–atmosphere interactions at regional and global scales.

## Discussion
We detect a systematic impact of extensive irrigation at the global scale on the location and downwind rainfall amount of afternoon rain within a 50-km radius of irrigated areas. We deliberately chose a simplistic approach based on monthly wind direction climatologies and a straightforward detection algorithm to select rain events, which, in light of all the possible limitations, still enables us to detect a robust signal. Nevertheless, variability and uncertainty in our results can occur due to a wealth of potential factors. Using more specific data and improved event detection can potentially provide a better selection of rain events. For example, due to the lack of available data, there is no specific information on the timing of irrigation at particular grid cells. While we assume irrigation occurs only when the monthly mean temperature exceeds 10 °C, we still detect many events where nearby irrigated grid cells were likely not irrigated at that time. This further suggests that part of the signal we find may not solely originate from irrigation but possibly also from variations in land cover. The selection of rain events requires the implementation of several assumptions, e.g. concerning the choice of the irrigation threshold (AEI >80%). When lowering the irrigation threshold, i.e. including rain events in regions less equipped for irrigation, the detected downwind signal vanishes (see Fig. 6). On the other hand, considering an even higher threshold (AEI >90%), the signal becomes even more pronounced. This, in fact, strengthens our hypothesis that extensive irrigation or associated land-use changes are influencing afternoon rain across downwind areas. On the other hand, lowering the irrigation threshold has a minimal effect on the upwind signal, suggesting that the upwind signal may occur due to specific features in the sample of detected afternoon rain events or is influenced by different mechanisms potentially linked to land-use changes rather than direct irrigation activities. In conclusion, these results indicate that large-scale irrigation activities mainly influence the downwind signal. We detect a peak in afternoon rainfall likelihood of approximately three grid cells downwind of irrigated

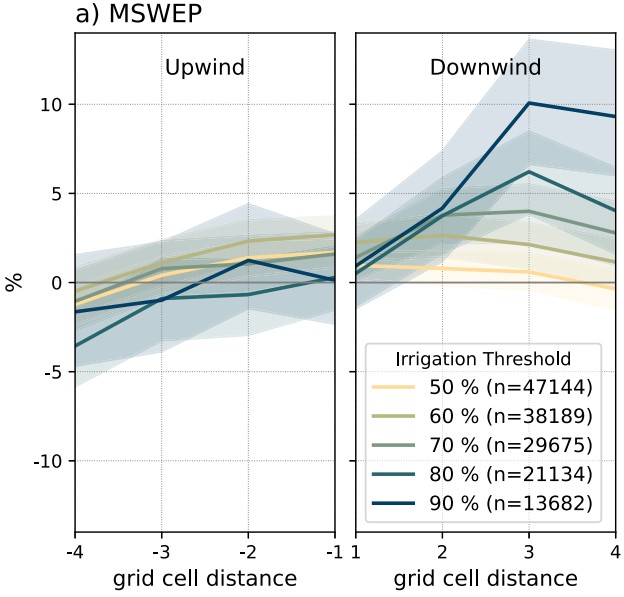

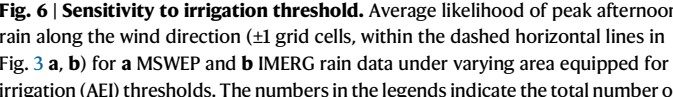

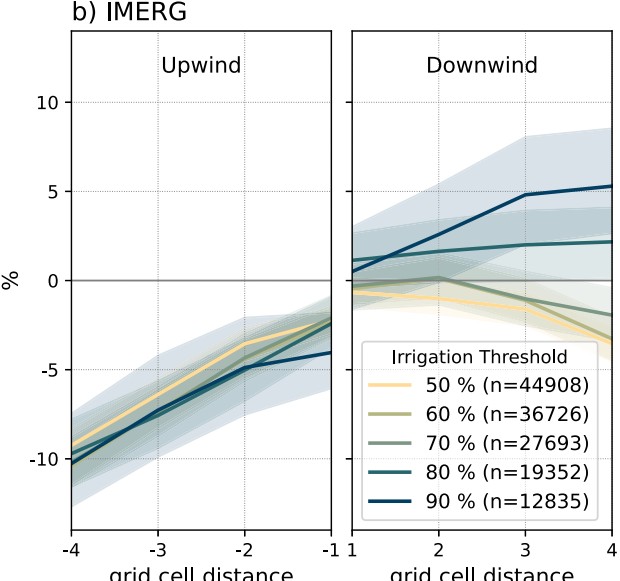

**Fig. 6 | Sensitivity to irrigation threshold.** Average likelihood of peak afternoon rain along the wind direction (±1 grid cells, within the dashed horizontal lines in Fig. 3 **a**, **b**) for **a** MSWEP and **b** IMERG rain data under varying area equipped for irrigation (AEI) thresholds. The numbers in the legends indicate the total number of detected rain events for each irrigation threshold. The shaded area represents the respective 80% confidence interval (CI) based on a bootstrapping approach (see Methods).

areas for MSWEP and about four grid cells downwind for IMERG (see Fig. S6). This indicates a clear local irrigation–atmosphere feedback that affects the occurrence of afternoon rain within 50 km of irrigated areas. However, varying land surface and atmospheric characteristics may alter the peak location. For example, while the peak is evident in some regions (e.g. North America and East Asia), it may be found outside the considered 50-km radius in other areas (see Fig. S4).

We further implement various other thresholds. The choice of the afternoon rain threshold, maximum elevation difference and temperature threshold only slightly affects our results (see Fig. S7). We are confident in detecting signals associated with afternoon rain, as the signal is less pronounced when considering no morning rain threshold (see Fig. S8). However, while we account for various factors to isolate afternoon rain events caused by local meteorological conditions and non-orographic convection, some of the detected events may still be influenced by large-scale advection, distant orographic effects, or nearby water bodies. For example, diurnal wind variations between low-lying irrigated areas and mountain ranges may still dominate regional precipitation dynamics beyond the 50-km scale and the afternoon period considered in this study[30]. Additionally, when considering monthly wind climatologies, daily wind directions and the associated upwind and downwind patterns for individual rain events may vary. As a consequence, part of the gradient in the upwind signal might occur due to afternoon wind direction shifts, resulting in a higher likelihood of afternoon rainfall closer to or at irrigated grid cells.

Despite the range of potential limitations, we find robust signals on irrigation impacting afternoon rainfall. We highlight the robustness of our findings by using Monte Carlo and bootstrapping techniques. Both precipitation datasets agree that precipitation is more likely downwind of irrigated grid cells (Figs. 3 and S3). However, there are discrepancies regarding the amount and likelihood of upwind precipitation. IMERG tends to locate peak afternoon rainfall more often directly at the location of the irrigated grid cells than MSWEP, which tends to detect more rain events downwind of irrigated grid cells. As we show relative percentage differences in Fig. 3, this leads to a more pronounced (positive) downwind signal in MSWEP and a more pronounced (negative) upwind signal in IMERG. Given the high-resolution

grid scale at which we assess rain events and irrigated grid cells, minor systematic spatial mismatches and temporal differences due to different data sources and processing of IMERG and MSWEP can constitute such a variation in the obtained signal.

Our findings offer large-scale observational evidence of how intensive irrigation may influence the likelihood and amount of afternoon rainfall. Assessing the obtained signal improves our understanding of the climate impact of irrigation on local-to-regional precipitation dynamics and land–atmosphere interactions and upscales previous regional findings on irrigation–atmosphere processes. Our results support recent efforts of the modelling community to implement irrigation schemes in (regional) climate models and can help to improve and constrain the modelled irrigation–atmosphere coupling by providing much-needed benchmark information beyond regional scales. This is particularly valuable for enhancing rainfall and storm forecasts and climate predictions over and near heavily irrigated areas, leading to a better understanding of coupled land–atmosphere processes and aiding water resources management and planning in these regions.

## Methods

### Precipitation data

We use two global, high resolution (0.1°), sub-daily precipitation datasets: (i) Multi-source weighted-ensemble precipitation (MSWEP) v2.8 3-hourly precipitation[42,43] and (ii) the integrated multi-satellitE retrievals for GPM (IMERG) V07 half-hourly precipitation dataset[44]. Both datasets are available during the 2001–2020 study period. MSWEP merges multiple rain gauge databases, satellite observations and reanalysis data to obtain high-quality global precipitation estimates. The data undergoes distributional bias correction and corrects systematic errors across the land surface using streamflow observations. IMERG is based on passive microwave remote sensing data obtained as part of the global precipitation measurement (GPM) mission. It uses the GPM core observatory satellite as the standard, combined with observed precipitation estimates from various other satellites and infused with monthly gauge information. Despite differences between the underlying data and the data processing, both datasets demonstrate a similarly strong performance in the

contiguous United States with only small biases[43], therefore providing a solid foundation for our analyses. Nonetheless, we find distinct differences between the rainfall distributions of MSWEP and IMERG, with MSWEP generally showing larger rain amounts except for the 99th percentile (see Supplementary Table S1). These differences may arise from fundamental differences in the methodologies used to estimate rainfall (see above), which can lead to systematic differences in the precipitation distributions.

**Afternoon rain event detection criteria**
We define afternoon rain as the total amount of rain between 12- and 24-h local time. At each longitude, we determine the closest 3-hourly (MSWEP) and half-hourly (IMERG) time step (in UTC) to the actual solar noon estimated using the formula 12 − lon/15° for longitudes lon ranging between −180° and 180°. Due to the 3-hourly (MSWEP) and half-hourly time-stepping (IMERG), afternoon periods may not necessarily reflect solar hours from 12 to 24 h as they could be subject to the respective delays (up to 3 h for MSWEP, and up to 30 min for IMERG). We intentionally consider a long (half-day) afternoon period to account for regional differences in the timing of afternoon rainfall. Please note that considering a shorter afternoon period (12 to 18 h) reveals a similar signal, although it is less robust (Fig. S9).

Different criteria are used to select individual afternoon rain events most likely triggered by local meteorological conditions instead of other factors, such as synoptic-scale and orographic mechanisms (see also Fig. 1). We only consider relatively strong afternoon rain events with local peak rain above 3 mm[36] (see Fig. S7 for a sensitivity analysis on the choice of this threshold). To detect rain events actually triggered in the afternoon, we exclude afternoon events succeeding morning rain (0 to 12 h) above 1 mm (see Fig. S8 for results obtained without the morning rain threshold).

**Irrigation influences.** We identify rain events near irrigated areas using straightforward selection criteria. We select rain events featuring at least one irrigated and one non-irrigated grid cell within a 50-km radius surrounding the peak location. That means, multiple irrigated grid cells can be attributed to individual afternoon rain events. However, ca. 75% (MSWEP) and 70% (IMERG) of all detected rain events feature less than 20% irrigated grid cells within the 50-km radius surrounding the peak location (see Fig. S5). Imposing a certain minimum percentage of irrigated/non-irrigated grid cells instead may lead to spurious results due to the clustered nature of irrigated areas. Irrigated areas are identified based on the Food and Agriculture Organization (FAO) global map of irrigated areas[45]. The data provides a high-resolution (5′) map of grid cell AEI around the year 2005 in the percentage of the total area per grid cell. We consider individual grid cells as being irrigated if more than 80% of the grid cell area is equipped for irrigation (see Fig. 6 for a sensitivity analysis on the choice of this threshold). Using two other irrigation data sources providing (i) updated AEI for the year 2015[48] and (ii) extended data[49] showing 18% more irrigated areas worldwide in comparison to the FAO data used within this study, yields a weaker signal, particularly when considering the extended data. However, it is important to note that the extended data also includes more widespread sparsely irrigated regions. Rain events identified near these areas are likely not directly influenced by large-scale irrigation activities, which contaminate the overall signal (see Fig. S10). We estimate regional fractions of irrigated areas relative to the total cropland area (see Fig. 2) using moderate resolution imaging spectroradiometer (MODIS) land cover climate modelling grid (CMG) (MCD12C1) version 6.1 data[50].

**Topographic influences.** We use ETOPO 2022 elevation data[51] to only select rain events with minor elevation differences (<300 m) between all grid cells within a 50-km radius surrounding the peak location (see Fig. 1). ETOPO 2022 includes bathymetry data, which enables us to identify ocean grid cells (<0 m) to exclude rain events potentially affected by coastal dynamics. Both criteria ensure that we exclude mountainous and coastal regions. The elevation data (native resolution: 1′) has been regridded to 0.1° using bilinear interpolation. Therefore, we may average out small-scale elevation features. However, the choice of the elevation threshold only marginally affects the detected signal (see Fig. S7).

**Growing seasons.** Global geospatial information on the specific timing of irrigation is lacking. However, we assume that irrigation is only applied during the crop growing season under conditions that constitute an irrigation demand (e.g. plant water stress). As a proxy for crop growing seasons, we set a temperature threshold of monthly mean temperatures at 10 °C (see Fig. S7 for a sensitivity analysis on the choice of this threshold). For this purpose, we use climatic research unit gridded time series (CRU-TS) 0.5° mean monthly temperature data[52] to compute monthly temperature climatologies from 2001 to 2020. Temperature data has been regridded to 0.1° using bilinear interpolation.

We chose a 50-km radius surrounding peak afternoon rain locations, which is twice the typical size of afternoon rain events and thunderstorms (roughly 25 km[53]). This radius also corresponds to the typical length scales considered in field campaigns[17,21]. The resulting diameter of 100 km provides a sufficient distance to detect a robust signal, while also including a large number of afternoon rain events. Considering a larger radius drastically reduces the number of detected rain events, while a similar signal can still be detected within a 70-km radius (see Fig. S6). This analysis, considering a 70-km radius, demonstrates a distinct decline in downwind impact beyond 30 to 40 km and almost no measurable signal at distances of 50 to 60 km downwind. Areas surrounding peak locations are allowed to overlap. Approximately 13 (MSWEP) and 11% (IMERG) of all detected rain events partly overlap, i.e. an irrigated grid cell can be within 50 km of two separate rain events. Since these irrigated grid cells may influence the formation of both rain events, we have decided to keep overlapping events in our analysis. However, excluding these events does not significantly impact our results (see Fig. S11).

**Wind-based reference system**
For each rain event, we determine the position of irrigated grid cells relative to the peak afternoon rainfall location with respect to wind direction. Due to the lack of satellite-based wind data over land surfaces and within the study period, we use ERA5 10-m wind data[46] to compute a monthly wind direction climatology from 2001 to 2020. ERA5 provides zonal and meridional wind speed data at 0.25°, which we regridded to 0.1° using nearest neighbour methods to compute the wind direction at each grid cell. Please note again that we refrain from using daily or sub-daily wind directions obtained from ERA5. However, we provide results based on monthly wind data (instead of wind climatologies) in the Supplementary Information (see Fig. S12). To define up- and downwind areas, we rotate the irrigation data surrounding the peak location for each event based on the wind direction (using scipy.ndimage.rotate[54]). This procedure enables us to define a common reference system based on the respective wind direction for all rain events. The centre location of each reference system corresponds to the location of the peak afternoon rainfall, whereas the wind direction is directed to the right.

**Afternoon rain likelihood**
We compute the total count of irrigated grid cells at each grid location using the wind-based reference system for each event. Within the reference system centred around the peak afternoon rain locations (see Fig. S13), grid locations to the left of the centre define areas upwind of peak afternoon rainfall, i.e. the peak location is located downwind of these areas. Similarly, grid locations to the right of the

centre define areas downwind of peak afternoon rainfall, i.e. the peak rain location is located upwind of these areas. To make this more intuitive, as presented in the main text, we transform the wind-based reference system such that the centre location represents the location of individual irrigated grid cells. The transformation toward an irrigation-centred reference system can be realised by rotating the rainfall-centred reference system by 180° equivalent to a point reflection over the origin (centre point). In the transformed reference system, grid locations to the left (right) of the centre define areas upwind (downwind) of the respective irrigated grid cell. Hence, each grid location provides the total count of rain events centred around irrigated grid cells. In the figures presented in the main text, we assess the likelihood of peak afternoon rainfall relative to the centre location in terms of a relative percentage difference (RPD) as follows:

$$\text{RPD} = 100 \times \frac{n_{i,j} - n_{0,0}}{n_{0,0}} \qquad (1)$$

with $n_{0,0}$ denoting the number of rain events at the centre location and $n_{i,j}$ the number of rain events at the grid cell location $i, j$. We additionally isolate immediate up- and downwind areas as a function of grid cell distance from the centre by estimating the average peak afternoon rain likelihood in a band (±1 grid cell, i.e. three grid cells wide) to the left and right of the centre location (see Figs. 3c, d, 4). Please also note that we consider distances in terms of the number of grid cells. Actual distances in east-west directions (in km) may vary slightly depending on latitude. However, these differences are minimal since most irrigation activity occurs in tropical and subtropical latitudes.

### Uncertainty assessment

Within the band directly up-/downwind of the centre location, we estimate the uncertainty of peak afternoon rainfall likelihood in a bootstrapping approach. We randomly select (1000 times, with replacement) $n$ events (with $n$ representing the total number of rain events) and compute the average likelihood as a function of grid cell distance from the centre (see dark shading representing the 80% confidence interval in Figs. 3c, d, 4). We further estimate an expected value of peak afternoon rainfall likelihood in a Monte Carlo approach by randomly sampling wind direction (within the range from 0° to 360°) for each rain event 1000 times (see the dashed line representing the average and pale shading representing the 80% confidence interval in Figs. 3c, d, 4). By doing so, we can isolate the signal from the actual wind direction data against the null hypothesis that wind direction near irrigated grid cells does not affect the likelihood of afternoon rain up or downwind of irrigated grid cells.

### Subsampling regional signals

To assess whether conditions in South Asia dominate the overall signal (see Fig. 3), we isolate all rain events occurring in South Asia against those events occurring outside (see Fig. 4a, b). South Asia is delineated based on the current Intergovernmental Panel on Climate Change (IPCC) definition of climate regions[55] (see also Fig. S1). A total number of $n$ = 9630 (MSWEP) and $n$ = 7437 (IMERG) rain events occur in South Asia, while $n$ = 10,323 (MSWEP) and $n$ = 11,822 (IMERG) rain events have been detected outside South Asia.

To assess whether the detected signals may differ for irrigated regions under more arid vs. more humid climatic conditions, we equally part the respective samples for MSWEP and IMERG considering the aridity index (AI). We use AI estimates from the Global ET0 and Aridity Index Database Version 3[56], providing high-resolution (30˜) geospatial data of averaged aridity conditions for the 1970–2000 period. While this dataset precedes our study period (2001–2020), AI is a climatological feature representing historical conditions of dryness and wetness and is used here for classification purposes. We regrid the AI data to our 0.1° target grid using bilinear interpolation. We sort the

full samples of rain events (for each MSWEP and IMERG) according to their climatological AI value at the peak rainfall location of each event and split the sorted samples into two halves, thus representing the most arid and humid regions. For MSWEP, the most arid (humid) half corresponds to aridity conditions of AI <0.45 (AI >0.45), whereas for IMERG it corresponds to AI <0.39 (AI >0.39). The most arid half represents rain events occurring under arid to semi-arid conditions, while the most humid half represents events occurring under semi-arid, dry sub-humid and humid conditions. Please refer to Fig. S14 for the respective results in arid (AI <0.2), semi-arid (0.2 < AI <0.5) and humid (AI >0.5) regions. We detect the strongest downwind signal under arid conditions, whereas the downwind signal becomes slightly negative under humid conditions.

### Quantile-quantile mapping of rainfall amounts

We assess afternoon rainfall amounts of events occurring downwind vs. non-downwind of irrigated grid cells. We consider the total amount of rain between 12 and 24 h. To determine whether downwind vs. non-downwind rain events differ in their rainfall amount, we use quantile-quantile plots (Q-Q plots, see Fig. 5). Q-Q plots provide a non-parametric approach to compare the shape of two sample distributions by plotting their quantiles against each other. We define downwind rain events as those events that occur in the downwind band of irrigated grid cells (see e.g. Fig. 3c, d), i.e. all events that feature at least one irrigated grid cell upwind of their peak location (±1 grid cell) or directly at the peak location. In Fig. 5, we consider all events with no irrigation at the peak location and outside the downwind band as non-downwind rain events. However, please refer to Fig. S15 for Q-Q plots only considering those events as non-downwind events that are outside the entire downwind half. As we detect more downwind rain events (MSWEP: $n_{\text{down}}$ = 10,691, $n_{\text{up}}$ = 10,443; IMERG: $n_{\text{down}}$ = 10,383, $n_{\text{up}}$ = 8969), we use an interpolated quantile estimate to enable comparability of the distributions. We assess uncertainties using a Monte Carlo approach. We randomly select two samples of sizes $n_{\text{down}}$ and $n_{\text{up}}$ from the full sample (1000 times) and generate the associated Q-Q plots without imposing any further assumption on wind direction. From the ensemble of randomly sampled Q-Q plots, we estimate the IQR, 80 and 95% confidence interval (shading in Fig. 5)

### Data availability

The MSWEP precipitation data are available at https://www.gloh2o.org/mswep/. The IMERG precipitation data are available at https://disc.gsfc.nasa.gov/datasets/GPM_3IMERGHH_07/summary. The FAO Global Map of Irrigated Areas is available at https://www.fao.org/aquastat/en/geospatial-information/global-maps-irrigated-areas. The two alternative irrigation datasets are available at https://zenodo.org/records/7809342. ERA5 reanalysis data are available at https://cds.climate.copernicus.eu/datasets/reanalysis-era5-complete. The ETOPO 2022 elevation data are available at https://www.ncei.noaa.gov/products/etopo-global-relief-model. The CRU temperature data are available at https://crudata.uea.ac.uk/cru/data/hrg/. The IPCC region delineation are available at https://github.com/SantanderMetGroup/ATLAS. The Aridity Index data are available at https://doi.org/10.6084/m9.figshare.7504448.v5. MODIS land cover data are available at https://doi.org/10.5067/MODIS/MCD12C1.061. The minimum data required to reproduce the figures are available at https://doi.org/10.5281/zenodo.14028492.

### Code availability

Data analysis was performed using Python 3.11.4. We used Jupyterhub (https://jupyter.org/hub) within the Jupyterhub @ DKRZ environment (https://jupyterhub.dkrz.de) provided by the German Climate Computing Centre (DKRZ, https://www.dkrz.de). Python scripts and Jupyter notebooks for the analysis are available at https://doi.org/10.5281/zenodo.14028492. Unless otherwise indicated, data pre-processing was carried out using default Climate Data Operators (cdo) functions[57].

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

## Acknowledgements

This research was carried out in the UWaRes project supported by the Helmholtz Association Initiative and Networking Fund. This work used resources of the Deutsches Klimarechenzentrum (DKRZ, https://www.dkrz.de) granted by its Scientific Steering Committee (WLA) under project ID ch0636. We thank Stefan Kern and the Integrated Climate Data Center (ICDC https://www.cen.uni-hamburg.de/icdc/), CEN, University of Hamburg, Hamburg, Germany, for data support. P.G. and A.U.S. thank Katharina Bülow for providing an internal review.

## Author contributions

P.G. conceived and designed the study and performed the data analysis. A.U.S. and D.G.M. provided substantial input to finalise the analysis. S.M. assisted in the selection of irrigation data. K.L.F. provided significant input to the discussion of the results, and A.G.-G. and J.P. helped in the overall interpretation of the results. P.G. and A.U.S. drafted the paper. D.G.M., S.M., K.L.F., A.G.-G. and J.P. provided additional references and discussed and revised the paper in several iterations.

## Funding

## Competing interests

The authors declare no competing interests.
