## [Transparent Peer Review file · Nature Communications]

Observational Evidence of Increased Afternoon Rainfall Downwind of Irrigated Areas

Corresponding Author: Dr Peter Greve

Version 0:

Reviewer comments:

Reviewer #1

(Remarks to the Author)

The paper used observational data to show that afternoon rain events occurs more often downwind of extensively irrigated land area. Although there have been some regional analyses on this, a global analysis could provide more detailed information. However, this should be strengthened in the paper (see below). Overall, I think the paper could be considered for publication in NC after some major revision.

1. There have been some observational studies on the impact of irrigation on precipitation (e.g. Alter et al.), the difference of this study compared to previous studies should be emphasized (e.g., large-scale and global).
2. I suggest the authors to perform some regional analysis and show the differences in precipitation response to irrigation, such as in Fig. 3. Currently, the paper only shows South Asia and other regions. It is interesting to see results in more irrigation regions. After all, a major advantage of this study compared to previous studies is the spatial coverage.
3. Some regions have strong diurnal variation of wind directions, where the wind directions may be opposite during the day and night. For example, in North China Plain where the mountain-plains solenoid strongly controls the diurnal variation of winds and precipitation. The preferred time of precipitation also varies in different regions. These may be hard to consider in your analysis, but you can at least mentioned it.
4. Fig. 3. Please be clear how the values in (a)(b)—the likelihood of peak location of afternoon rain relative to irrigated grid cells—are calculated. I can't find it in main text or Methods.
5. The Q-Q plots indicate that the upwind sample distribution is more dispersed. Does it mean the distribution of upwind precipitation has longer tails (more extreme values)? How could this be related to irrigation? The paper does not discuss it in detail.
6. There are some previous work closely related to this study but are not mentioned.
 - (i) Contribution of irrigation to local and remote precipitation based on moisture tracking: Wei, J., P. A. Dirmeyer, D. Wisser et al., (2013) Where does the irrigation water go? An estimate of the contribution of irrigation to precipitation using MERRA, *J. Hydrometeorology*, 14, 275-289.
 - (ii) Impact of irrigation on the diurnal cycle of precipitation: Song, Y. et al. (2024) Irrigation in the North China plain regulates the diurnal cycle of precipitation and regional water cycle. *Climate Dynamics*. <https://doi.org/10.1007/s00382-024-07208-z>

Other minor comments:

Line 75. corresponding with ->corresponding to?

Line 97-102. "Our findings offer ..." The significance of the study should be moved to the end of the paper.

Line 138. Why directly at the location of irrigated grid cells? Fig.3 does not show that.

Line 144. "larger signal using MSWEP". Please be clear.

Line 289-291. "Due to the 3-hourly (MSWEP) and half-hourly time-stepping (IMERG) ...could be subject to respective delays." Half-hourly data still subject to delays?

Line 320. "elevation differences (<300m) within a 50 km radius." Is 300m the maximum difference between grid cells?

Figure 2. The shading of area equipped for irrigation is covered by the rain events (yellow dots) and is not clearly shown.

(Remarks on code availability)

Reviewer #2

(Remarks to the Author)

General remarks

This manuscript provides observational evidence that afternoon rainfall more frequently occurs downwind of irrigated areas than over or upwind. This evidence was provided by two global, high-resolution rainfall products at 0.1-degree resolution and various other geospatial datasets. However, rainfall intensity seems to generally be lower downwind of irrigated areas, especially for higher percentiles of intensity. The authors offer some potential mechanistic explanations for these observed signals and provide supplemental evidence of these signals by showing figures that vary the thresholds chosen in the manuscript.

Overall, I found this manuscript to be very well-written and generally sound in its methodology. In my opinion, the findings are novel and are important for the scientific community to consider as weather and climate models continually improve their spatiotemporal resolutions and the representation of land-atmosphere interactions. I do have some comments about the figures, the methodology, and the robustness of some of the findings, but I think that if they are successfully addressed, this manuscript will have good potential for ultimate publication in Nature Communications.

Major comments

Figure 1 – Overall, I think this is a good figure, but I have several comments about it:

- I would change the "x" in the center of the middle and right panels to be a different shape and/or different color, since it's very difficult to see the "x" in the right panel and it would help to differentiate it from the "x's" associated with the irrigated grid cells.
- I can tell apart the irrigation colors from the rain colors in the figure, but they're similar enough that I think other readers may get confused. I would recommend switching the colorbar for one of these variables to enable a better contrast.
- In step 4 of the flow chart, I would add "both" before the word "irrigated". The reason for this is that when I first read this text, I wasn't sure why it was important to mention "irrigated" and "non-irrigated" in this bubble, since those are the only grid cell options. After reading the Methods, I see that the presence of *both* irrigated and non-irrigated grid cells in the same area of interest is required, but it's not clear just from reading this figure.
- In the US, "exemplary" can be taken to mean "excellent", so seeing this word here was very confusing to me. I would recommend removing or rephrasing this word.
- I understand how this rain event-centered grid works, but it's a bit of a mental jump for me to go from this system to the irrigation-centered system in Figure 2. Since the system in Figure 2 is more intuitive to me, I think it could be beneficial to see if Figure 1 can be reframed in the irrigation-centered system.

Figure 2 – The caption says that "Highlighted regions indicate areas where afternoon rain events (see Fig. 1 and Methods) occur close (< 50km) to irrigated grid cells..." However, it seems from the methodology that rain events are actually considered "over" irrigated grid cells as well, and in the figure itself, there are many yellow areas seemingly overlapping irrigated areas. If this is true, then I would say in the caption "over and close (< 50 km) to". Also, this is optional, but it could be helpful to see if there is a way to separately indicate irrigated areas and rain events (i.e., prevent the yellow rain events from covering the irrigation colors).

Figure 3 – It actually seems that the upwind (negative) effect is greater than the downwind (positive) effect for IMERG. Any thoughts as to why this may be occurring?

Lines 172-174 – It's very difficult to see a difference in MSWEP around the 90th percentile in Figure 5. It's more apparent higher than the 90th percentile, so I would change this text to say "(above Q90)" instead of "(around Q90)".

Lines 179-181 – I wouldn't say that both Q-Q plots "clearly" emerge from the envelope of uncertainty. The MSWEP plot seems to be around the edge of the 95% CI, so the emergence isn't as clear. I would change this to say "both Q-Q plots more clearly emerge" instead.

Line 181 – It could be argued that the impact on afternoon rain is “upwind” as well as “downwind”, since Figure 5 is just comparing the relative difference between the two regions. Because of this lack of conclusiveness, it might be more accurate to say “suggest a distinct local impact on the total amount of afternoon rain” or something similar.

Figure 5 – Any thoughts as to why IMERG shows a much larger separation from MSWEP in the higher percentiles? Could it be related to IMERG’s much finer temporal resolution? Also, it would be helpful to add a description of the solid black 1:1 line in the figure caption, and it might be helpful to add gray dots to the figure legend.

Lines 193-198 – I see that the focus is on the downwind pattern, but I think it’s worth considering the upwind pattern more closely. For example, IMERG seems to encounter even larger changes upwind than downwind, for likelihood (Figure 2) and possibly for intensity (Figure 5). In fact, the decreases in likelihood upwind for IMERG are stronger than the increases in likelihood downwind for MSWEP (Figure 2). Therefore, it may be helpful to add more text discussing this phenomenon, since it may be on par with the downwind pattern. It may also be worth considering the *difference* between upwind and downwind likelihoods as part of this study, since the difference between downwind and upwind seems similar between MSWEP and IMERG in Figures 2 and S2 (and other figures) even though the actual values are different.

Figure 6 – I noticed in this figure that the upwind values are rather insensitive to the irrigation threshold. Does that mean that upwind areas, which have strong decreases in likelihood for IMERG, are not affected by irrigation? Or does it mean that the relationship is related to irrigation but to a different mechanism than the downwind area (like the one mentioned in lines 221-222)? There are no conclusive answers here, but I think it could be worth briefly discussing this in the manuscript.

Lines 291-292 – I think changing from a 12-hr to 6-hr period does have a noticeable change on MSWEP in the downwind region. While the shading is greater than zero for the 2-4 grid cell range from 12-24 hr, it only seems to be greater than zero for the 4 grid cell value from 12-18 hr. So there still seems to be an effect, but it seems less robust. It would be worth mentioning this in the manuscript.

Lines 314-316 – Actually, the downwind signal effectively disappears when switching to the MEIER-AEI dataset. Interestingly, the IMERG upwind signal strengthens, too. Any thoughts as to why this is the case?

Lines 339-340 – How much do the detected rain events decrease with larger radii? If feasible, it could be telling to analyze whether these patterns hold with larger radii, even 60 or 70 km.

Lines 420-423 – I’m a little confused about the methodology for choosing the “downwind” and “upwind” areas for Figure 5. It seems that “downwind” only refers to the downwind “band” (12 grid cells) whereas almost everything else is considered “upwind”. I may be misunderstanding this, but if my understanding is correct, then the regions considered would be very different sizes. Even for Fig. S11, it seems that half of the area of interest would be considered upwind but only the downwind “band” would be considered “downwind”. Any clarification would be much appreciated.

Figure S9 – I think it’s important to better differentiate in the figure caption how the “mean monthly wind data” in this figure differs from that of the original data (i.e., the climatology).

Minor comments

Line 20 – “Yet, the effect on rainfall patterns near irrigated areas remains less clear” – compared to what?

Lines 131-133 – I understand the rationale for focusing on monthly winds, but out of curiosity, were sub-daily winds actually used within this methodology (even if it’s not ultimately shown in the manuscript)?

Lines 137-139 – This sentence is a bit confusing to me. It might be more straightforward to say that the likelihood of peak afternoon rain directly at the location of the irrigated grid cells is between the likelihoods upwind and downwind.

Line 164 – I think the figure reference “(see Fig. 4 c,d)” belongs at the end of the previous sentence, because that sentence directly refers to the results in the figure.

Figure 4 – Any thoughts as to why there is a stronger signal closer to the irrigated grid cells in the rest of the world compared to South Asia (at least for MSWEP)? Also, any thoughts as to why MSWEP is showing increased likelihood both downwind *and* upwind in South Asia?

Lines 289-291 – Is the precipitation distribution for MSWEP similar to that of IMERG for the rain events used in this study? I’m curious whether the shorter time step for IMERG results in a different distribution.

Lines 295-296 – Was there a particular reason for initially choosing 3 mm as the threshold value?

Lines 303-305 – One thing to consider: Even if the total irrigated grid cell percentage within 50-km of a rain event is <20%, it’s possible that many grid cells upwind of the center location are also irrigated. Perhaps that’s related to the strong decreases in likelihood in IMERG.

Lines 401-403 – The aridity dataset seems to only cover 1970-2000, but the MSWEP and IMERG datasets cover 2001-2020.

Is it possible that aridity experienced major changes since 2000 that would majorly affect these results?

Line 404 – Should “targeted” be “target”?

Lines 409-411 – Since it seems that the two current aridity categories overlap for “semi-arid”, I wonder if it would be helpful to conduct the analysis with three aridity categories that more cleanly cover the different aridity conditions. If such an analysis provides useful results, the results could be substituted for Figure 4 or provided as a new supplementary figure.

Lines 445-447 – Regarding the code: The title on the Zenodo page doesn't match the title of the manuscript – I'm not sure if that's intentional. Also, the files are Restricted, so I can't actually view the code files. I don't think seeing the code is essential for this review, but I hope that the code will be made available by the time of publication.

Line 533 – There seems to be a typo on this line.

Figure S10 – The “x” in the center of panel b) is almost invisible. It would be probably helpful to change its color (and perhaps that of panel a)) to improve the contrast.

(Remarks on code availability)

Version 1:

Reviewer comments:

Reviewer #1

(Remarks to the Author)

The authors have addressed most of my comments. I have one more comment for the authors to consider.

As can be seen in Figs. 3 and 4, the mean peak location of afternoon rain relative to irrigated grid cells is at around 3 grid cell distance. Thus, my question is what determines the distance and how the distance changes in different regions/occasions. Possible factors include wind speed, surface aridity, and topography. Hope the authors can provide some insight or analysis on this.

(Remarks on code availability)

Reviewer #2

(Remarks to the Author)

I very much appreciate the authors comprehensively addressing all of the comments from my previous review. I believe that this manuscript is nearly ready for acceptance into Nature Communications. Following are some minor additional comments:

Line 190 - I think it would be helpful to be consistent with the terminology regarding the 12-cell downwind "band" that is used for Figure 5 (and Figure 3c,d), especially since this term is already used in the Methods section (e.g., line 451). I would change "section" to "band" in this line and add the term "upwind and downwind bands" somewhere within this parenthetical expression in the caption of Figure 3: "(+/-1 grid cells, within the dashed horizontal lines in a, b)".

Figure 5 - Please change "Upwind Rain" to "Non-Downwind Rain" for the y-axis label and change "downwind section" to "downwind band" in the figure caption.

Lines 294-295 - I see that the word "upwind" was added here, but I'm a little confused because the "signal is less pronounced" only for the downwind band. Any clarification would be much appreciated.

Lines 401-402 - Actually, the signal seems to go slightly negative (not just weaker) for humid conditions. I would adjust this sentence to reflect this.

Figure S2 - Optional, but it might be helpful to change the color at the upper boundary of the colorbar to something that looks a little more different from the color of the water bodies in the map.

Figure S10 - I'm a little confused by the figure caption, because while it indicates that the 70-km data is dark green, it actually seems that the original (50-km) data is dark green. The 70-km data seems to be more of a dark gray color. It would be helpful to clarify this color in the caption.

(Remarks on code availability)

I successfully downloaded and opened all of the files that are posted to Zenodo. Though I only skimmed (and did not run) most of the files, it seems that the files would allow the results of the paper to be reproduced and would be a usable resource for

the community.

Reviewer #1 (Remarks to the Author):

The paper used observational data to show that afternoon rain events occurs more often downwind of extensively irrigated land area. Although there have been some regional analyses on this, a global analysis could provide more detailed information. However, this should be strengthened in the paper (see below). Overall, I think the paper could be considered for publication in NC after some major revision.

We sincerely thank the reviewer for their overall positive and thorough evaluation of the manuscript. We have carefully revised and improved the manuscript based on the comments and suggestions and are convinced that the manuscript now better highlights the novel aspects of our assessment. Below, you will find our detailed responses to the individual comments.

1. There have been some observational studies on the impact of irrigation on precipitation (e.g. Alter et al.), the difference of this study compared to previous studies should be emphasized (e.g., large-scale and global).

We thank the reviewer for suggesting a more thorough comparison of our findings to previous studies on irrigation–precipitation interactions. We have revised the manuscript to better assess our results in relation to the previous research aiming to highlight the novelty of our work. We have adapted the text in the Results sub-section “Exploring Mechanisms in Irrigation–Rainfall Interactions” (l. 206-213 in manuscript with tracked changes) as follows:

This work shows that afternoon rain is more likely to occur downwind of irrigated grid cells globally, suggesting an influence of irrigation activities on regional precipitation patterns. Similar precipitation–irrigation interactions have been observed downwind of irrigated areas in regions such as the eastern Sahel (Alter et al., 2015a) and the Midwestern United States (Alter et al., 2015b). However, modelling experiments indicate a more varied response of local precipitation patterns to irrigation. These responses range from upwind (Im et al., 2014) to downwind rainfall increases (Alter et al., 2015b; Yang et al., 2017), and from less intense (Valmassoi et al., 2020; Lo et al., 2021; Whitesel et al., 2024) to enhanced local rainfall directly over irrigated areas (Wei et al., 2013).

2. I suggest the authors to perform some regional analysis and show the differences in precipitation response to irrigation, such as in Fig. 3. Currently, the paper only shows South Asia and other regions. It is interesting to see results in more irrigation regions. After all, a major advantage of this study compared to previous studies is the spatial coverage.

We thank the reviewer for suggesting a more thorough regional analysis. In response, we have included an additional Supplementary Figure S4 that illustrates the differences in precipitation to irrigation across various regions outside South Asia. These regions include North America, South America, Southern and Central Europe, Africa, East Asia, and Southern Australia.

However, we would like to stress that the overarching focus of our study is to provide a global perspective (as also noted by the reviewer) to better understand the universal mechanisms underlying irrigation–rain interactions. At the regional level, we expect substantial deviations from the global response identified in this study. Responses to irrigation at a regional level must be interpreted by accounting for local and regional meteorological, climatic, and non-climatic conditions (such as landscape properties), which we cannot address with sufficient depth given the global context of our study.

Figure S4: Average likelihood of peak afternoon rain along the wind direction normalized by randomized wind directions. Please note the different y-axes.

We still keep the the comparison of the regional response in South Asia against the detected signal in the rest of the world in the main text to highlight that the detected global signal is not predominantly influenced by mechanisms shaping irrigation–rainfall interactions in South Asia. South Asia features a large number of detected rain events, while the sample size of individual rain events is relatively low in many other regions. When assuming a 50km radius, we consider a total of 61 grid cells surrounding an irrigated grid cell. That means a small sample size (e.g., 500 detected rain events in a region) is distributed across these 61 grid cells, resulting in a relatively small count of rain events per grid cell. Consequently, even slight variations in the occurrence of rain events across the considered grid cells can result in significant (especially relative) differences and potentially mask the actual

response. For example, this may apply to the regional responses in low sample size regions such as South America and Africa. We urge the reviewer and the readers to interpret the obtained regional results with care.

Supplementary Figure S4 illustrates regional responses across six different regions across all continents. To account for the somewhat erratic behaviour when considering small sample sizes, we chose to present normalized results (considering randomized wind directions), as they better illustrate upwind vs. downwind differences. Regional responses across North America, South/Central Europe, and East Asia are consistent with the global signal. In other areas, such as South America, Northeast Africa, and Australia, the detected signals show some coherence with the global signal. However, the overall response in these regions remains generally inconclusive. It is important to note that these regions are characterized by smaller sample sizes (<1000 rain events), which results in larger associated uncertainties and different relative signals (please note the different y-axes). We recommend conducting a more detailed analysis at these regional levels, considering the different local-to-regional conditions shaping precipitation patterns and convection characteristics in specific locations.

We added the following text to the section on “Regional signals” (l. 163-176):

To assess whether conditions in South Asia dominate the overall signal, we divided the sample into rain events in South Asia and outside (see Fig. 4a,b). The comparison shows that similar signals are found within and outside South Asia, suggesting that the obtained global signal is not predominantly influenced by mechanisms shaping irrigation–rainfall interactions in South Asia. The decreased likelihood of peak afternoon rain upwind of irrigated grid cells is even more pronounced outside South Asia, especially when considering MSWEP data.

In individual regions outside South Asia, the sample sizes are much smaller, leading to significant uncertainties (Fig. S4). While regional responses across North America, South/Central Europe, and East Asia are consistent with the global signal, results in other regions remain inconclusive. These areas require more detailed assessments, explicitly considering local and regional meteorological, climatic and non-climatic conditions.

3. Some regions have strong diurnal variation of wind directions, where the wind directions may be opposite during the day and night. For example, in North China Plain where the mountain-plains solenoid strongly controls the diurnal variation of winds and precipitation. The preferred time of precipitation also varies in different regions. These may be hard to consider in your analysis, but you can at least mention it.

We thank the reviewer for that interesting insight into the regional atmospheric dynamics in the North China Plain. As correctly mentioned by the reviewer, accounting for these regional dynamics may be difficult at the global level considered in this study. We also agree with the reviewer that the timing of ‘afternoon’ rainfall may differ regionally. Therefore, we have decided to detect afternoon rain within the relatively long period between solar noon and midnight. We have added the following text to the Discussion (l. 296-301):

However, while we account for various factors to isolate afternoon rain events caused by local meteorological conditions and non-orographic convection, some of the detected events may still be influenced by large-scale advection, distant orographic effects, or nearby water bodies. For example, diurnal wind variations between low-lying irrigated areas and mountain ranges may still dominate regional precipitation dynamics beyond the 50km-scale and the afternoon period considered in this study (Song et al., 2024).

In the methods we now state (l. 355-356):

We intentionally consider a long (half-day) afternoon period to account for regional differences in the timing of afternoon rainfall.

4. Fig. 3. Please be clear how the values in (a)(b)—the likelihood of peak location of afternoon rain relative to irrigated grid cells—are calculated. I can't find it in main text or Methods.

We have added a more detailed explanation, including the respective equation (see below), of how we compute the likelihood of peak afternoon rainfall relative to the centre location in terms of relative percentage differences. Please refer to the Methods section on “Afternoon rain likelihood” in the updated manuscript (l. 444-449):

In the figures presented in the main text, we assess the likelihood of peak afternoon rainfall relative to the centre location in terms of a Relative Percentage Difference (RPD) as follows:

$$RPD = 100 \times \frac{n_{i,j} - n_{0,0}}{n_{0,0}}$$

with $n_{0,0}$ denoting the number of rain events at the centre location and $n_{i,j}$ the number of rain events at the grid cell location i, j .

5. The Q-Q plots indicate that the upwind sample distribution is more dispersed. Does it mean the distribution of upwind precipitation has longer tails (more extreme values)? How could this be related to irrigation? The paper does not discuss it in detail.

We thank the reviewer for this critical and correct observation. Our results suggest a more dispersed upwind sample distribution, indicating a longer tail and higher rainfall amounts during the most extreme events. We have clarified this in the Section on “Irrigation impact on total afternoon rain amount” as follows (l. 201-204):

Our findings, therefore, suggest a distinct downwind impact on the total amount of afternoon rain, indicating lower rainfall amounts during the most extreme events compared to those in other areas surrounding irrigated grid cells.

We can, however, only speculate on the potential impact of irrigation. As already mentioned in the Section on “Exploring Mechanisms in Irrigation-Rainfall Interactions”, a more in-depth assessment requires high-resolution and more detailed observations of near-surface wind patterns, atmospheric moisture, boundary layer characteristics, and land surface information. Many of these data are currently not available at the scales of this study. We have already provided a potential explanation for our findings, which we further extended in the revised version of the manuscript (l. 253-261):

One potential explanation for our findings is that moisture in combination with lower temperatures from upwind irrigated areas is more likely to initiate shallower convection, resulting in more frequent but less intense rain. At the same time, irrigation may lead to daytime mesoscale circulations, which can cause convergence upwind of irrigated areas (Lunel et al., 2024). Additionally, if convection is triggered in areas outside the cooler and moister downwind regions, it is more likely to happen under highly unstable atmospheric conditions, promoting deeper convection. These intense storms can create inflow patterns that draw in additional moisture from nearby irrigated areas, enhancing storm intensity.

6. There are some previous work closely related to this study but are not mentioned.

(i) Contribution of irrigation to local and remote precipitation based on moisture tracking: Wei, J., P. A. Dirmeyer, D. Wisser et al., (2013) Where does the irrigation water go? An estimate of the contribution of irrigation to precipitation using MERRA, *J. Hydrometeorology*, 14, 275-289.

(ii) Impact of irrigation on the diurnal cycle of precipitation: Song, Y. et al. (2024) Irrigation in the North China plain regulates the diurnal cycle of precipitation and regional water cycle. *Climate Dynamics*. <https://doi.org/10.1007/s00382-024-07208-z>

We thank the reviewer for pointing us to these very interesting and highly relevant publications, which are now referenced multiple times in the revised manuscript.

Other minor comments:

Line 75. corresponding with ->corresponding to?

Changed to (l. 78-80): *Rain intensities specifically increased between 1950 and 1980, coinciding with a rapid expansion of irrigation activities in the region.*

Line 97-102. "Our findings offer ..." The significance of the study should be moved to the end of the paper.

We removed that part. A similar paragraph is already featured at the very end of the paper.

Line 138. Why directly at the location of irrigated grid cells? Fig.3 does not show that.

Since the upwind areas show mostly negative relative results in Fig. 3 (especially when considering IMERG rain data), we conclude that there is a higher likelihood of afternoon rain directly at the location of the irrigated grid cell. We have rephrased the sentence as follows (l. 147-149):

This further indicates a likelihood of peak afternoon rain directly at the location of the irrigated grid cells which falls between the likelihood in downwind and upwind areas.

Line 144. "larger signal using MSWEP". Please be clear.

We rephrased that part to: *"... larger downwind signal using MSWEP, ..."*

Line 289-291. "Due to the 3-hourly (MSWEP) and half-hourly time-stepping (IMERG) ...could be subject to respective delays." Half-hourly data still subject to delays?

There can still be delays up to 30mins considering IMERG. We rephrased that sentence as follows (l. 352-354):

Due to the 3-hourly (MSWEP) and half-hourly time-stepping (IMERG), afternoon periods may not necessarily reflect solar hours from 12h to 24h as they could be subject to the respective delays (up to 3h for MSWEP, and up to 30min for IMERG).

Line 320. "elevation differences (<300m) within a 50 km radius." Is 300m the maximum difference between grid cells?

We acknowledge that the previous wording may be misleading. We have revised the text as follows (l. 388-390):

We use ETOPO 2022 elevation data to only select rain events with minor elevation differences (<300 m) between all grid cells within a 50km radius surrounding the peak location.

Figure 2. The shading of area equipped for irrigation is covered by the rain events (yellow dots) and is not clearly shown.

We thank the reviewer for bringing this to our attention. We decided to keep Fig. 2 as it is and include a clear version of the map showing AEI in the Supplementary Information (Fig. S2).

Figure S2: Global map showing the percentage of area equipped for irrigation (AEI) within each grid cell.

Reviewer #2 (Remarks to the Author):

General remarks

This manuscript provides observational evidence that afternoon rainfall more frequently occurs downwind of irrigated areas than over or upwind. This evidence was provided by two global, high-resolution rainfall products at 0.1-degree resolution and various other geospatial datasets. However, rainfall intensity seems to generally be lower downwind of irrigated areas, especially for higher percentiles of intensity. The authors offer some potential mechanistic explanations for these observed signals and provide supplemental evidence of these signals by showing figures that vary the thresholds chosen in the manuscript.

Overall, I found this manuscript to be very well-written and generally sound in its methodology. In my opinion, the findings are novel and are important for the scientific community to consider as weather and climate models continually improve their spatiotemporal resolutions and the representation of land-atmosphere interactions. I do have some comments about the figures, the methodology, and the robustness of some of the findings, but I think that if they are successfully addressed, this manuscript will have good potential for ultimate publication in Nature Communications.

We sincerely thank the reviewer for their overall positive and thorough evaluation of the manuscript. We have carefully revised and improved the manuscript based on the comments and suggestions

and are convinced that the manuscript now better strengthens the novel aspects of our assessment with improved figures, a better-explained methodology and an improved discussion of our findings. Please see below for our detailed response to the more specific comments.

Major comments

Figure 1 – Overall, I think this is a good figure, but I have several comments about it:

- I would change the “x” in the center of the middle and right panels to be a different shape and/or different color, since it’s very difficult to see the “x” in the right panel and it would help to differentiate it from the “x’s” associated with the irrigated grid cells.
- I can tell apart the irrigation colors from the rain colors in the figure, but they’re similar enough that I think other readers may get confused. I would recommend switching the colorbar for one of these variables to enable a better contrast.
- In step 4 of the flow chart, I would add “both” before the word “irrigated”. The reason for this is that when I first read this text, I wasn’t sure why it was important to mention “irrigated” and “non-irrigated” in this bubble, since those are the only grid cell options. After reading the Methods, I see that the presence of *both* irrigated and non-irrigated grid cells in the same area of interest is required, but it’s not clear just from reading this figure.
- In the US, “exemplary” can be taken to mean “excellent”, so seeing this word here was very confusing to me. I would recommend removing or rephrasing this word.

We thank the reviewer for the valuable comments. We have addressed all comments, resulting in a significant improvement of Fig. 1. We have, in particular, changed the “x” associated with irrigated grid cells to a drop icon. We have further changed the colour of the “x”, indicating the location of peak afternoon rainfall to make it more visible (and matching the colour of the 50km circle). We have further adapted the wording in Step 4 and the figure caption to avoid any confusion.

Figure 1: Illustration depicting the detection of an afternoon rain event featuring irrigated grid cells within a 50km radius surrounding the peak location.

- I understand how this rain event-centered grid works, but it's a bit of a mental jump for me to go from this system to the irrigation-centered system in Figure 2. Since the system in Figure 2 is more intuitive to me, I think it could be beneficial to see if Figure 1 can be reframed in the irrigation-centered system.

We fully understand the issue raised by the reviewer and considered reframing Fig. 1 toward an irrigation-centred system. However, we have finally decided to leave Fig. 1 as it is. The main reason for this decision is that the current version of Fig. 1 more accurately depicts the detection algorithm that was actually used to identify individual rain events. The transformation toward an irrigation-centred system is the final step in our methodology before visualizing the results. We have changed the description of Step 5 accordingly, which now reads: "Transformation toward a wind-based and irrigation-centred reference system". We further extended the figure caption, pointing the interested reader towards the Methods section, where the transformation is explained in more detail.

Figure 2 – The caption says that "Highlighted regions indicate areas where afternoon rain events (see Fig. 1 and Methods) occur close (< 50km) to irrigated grid cells..." However, it seems from the methodology that rain events are actually considered "over" irrigated grid cells as well, and in the figure itself, there are many yellow areas seemingly overlapping irrigated areas. If this is true, then I would say in the caption "over and close (< 50 km) to". Also, this is optional, but it could be helpful to see if there is a way to separately indicate irrigated areas and rain events (i.e., prevent the yellow rain events from covering the irrigation colors).

We agree with the reviewer that the previous figure caption could be misinterpreted. We have changed the text accordingly in the caption of Figure 2 and hope it avoids any potential confusion. We also thank the reviewer for pointing us towards the gold-coloured rain events covering the AEI estimates. We have decided to keep Fig. 2 as it is but providing a Supplementary Figure (Fig. S2, see above in our response to Reviewer 1) showing only AEI without the overlapping rain events.

Figure 3 – It actually seems that the upwind (negative) effect is greater than the downwind (positive) effect for IMERG. Any thoughts as to why this may be occurring?

As correctly mentioned by the reviewer, the upwind negative effect is more pronounced considering IMERG rain data, while the downwind positive effect is less strong. As we show relative percentage differences in Fig. 3 (which is now explained in more detail in the Methods section on "Afternoon rain likelihood"), this result indicates a larger fraction of detected afternoon rain events directly at the location of the irrigated grid cells in comparison to upwind areas. Hence, while both IMERG and MSWEP locate most events downwind of irrigated grid cells, it appears that IMERG tends to locate peak afternoon rainfall more often directly at the location of the irrigated grid cells, while MSWEP tends to locate rain events even more often downwind of irrigated grid cells. We can only speculate on potential reasons for this. However, given the high-resolution grid scale at which we assess rain events and irrigated grid cells, minor systematic mismatches between IMERG and MSWEP can constitute such a signal. We added the following text to the Discussion (l. 309-317 in manuscript with tracked changes):

However, there are discrepancies regarding the amount and likelihood of upwind precipitation. IMERG tends to locate peak afternoon rainfall more often directly at the location of the irrigated grid cells than MSWEP, which tends to detect more rain events downwind of irrigated grid cells. As we show relative percentage differences in Fig. 3, this leads to a more pronounced (positive) downwind signal in MSWEP and a more pronounced (negative) upwind signal in IMERG. Given the high-resolution grid scale at which we assess rain events and irrigated grid cells, minor systematic spatial

mismatches and temporal differences due to different data sources and processing of IMERG and MSWEP can constitute such a variation in the obtained signal.

Please also refer to our response to your comment below about the upwind signal being relatively insensitive to the choice of the irrigation threshold (see Fig. 6). The choice of the irrigation threshold systematically impacts the downwind signal, while the upwind signal shows no such sensitivity. This suggests that the upwind behaviour is not mainly driven by irrigation activities but may occur due to specific features in the sample of afternoon rain events or is influenced by different mechanisms potentially linked to land use changes rather than direct irrigation activities. In conclusion, our results indicate that large-scale irrigation mainly influences the downwind signal. We added the following text (l. 286-291):

On the other hand, lowering the irrigation threshold has a minimal effect on the upwind signal, suggesting that the upwind signal may occur due to specific features in the sample of detected afternoon rain events or is influenced by different mechanisms potentially linked to land use changes rather than direct irrigation activities. In conclusion, these results indicate that large-scale irrigation activities mainly influence the downwind signal.

Lines 172-174 – It's very difficult to see a difference in MSWEP around the 90th percentile in Figure 5. It's more apparent higher than the 90th percentile, so I would change this text to say "(above Q90)" instead of "(around Q90)".

We thank the reviewer for highlighting this important difference, which we have revised in the updated manuscript accordingly.

Lines 179-181 – I wouldn't say that both Q-Q plots "clearly" emerge from the envelope of uncertainty. The MSWEP plot seems to be around the edge of the 95% CI, so the emergence isn't as clear. I would change this to say "both Q-Q plots more clearly emerge" instead.

We revised the sentence accordingly.

Line 181 – It could be argued that the impact on afternoon rain is "upwind" as well as "downwind", since Figure 5 is just comparing the relative difference between the two regions. Because of this lack of conclusiveness, it might be more accurate to say "suggest a distinct local impact on the total amount of afternoon rain" or something similar.

We agree with the reviewer that our analysis does not clearly indicate whether the detected signal results from upwind or downwind impacts or a combination of both. We have revised the sentence accordingly (l. 201-204):

Our findings, therefore, suggest a distinct local impact on the total amount of afternoon rain, indicating lower downwind rainfall amounts during the most extreme events compared to those in other areas surrounding irrigated grid cells.

Figure 5 – Any thoughts as to why IMERG shows a much larger separation from MSWEP in the higher percentiles? Could it be related to IMERG's much finer temporal resolution? Also, it would be helpful to add a description of the solid black 1:1 line in the figure caption, and it might be helpful to add gray dots to the figure legend.

We have changed Fig. 5 accordingly and extended the figure legend, adding the grey dots. We further explain in more detail the meaning of the 1:1 line in the figure caption. We can only speculate about the clear separation in the higher percentiles for MSWEP and IMERG. It is important to mention that the bootstrapped uncertainty is larger using IMERG, especially for the higher quantiles. That indicates that the IMERG rain distribution is more dispersed in general. However, given the high-resolution grid and temporal scales at which we assess the data, minor systematic spatial mismatches and temporal differences due to different data sources and processing of IMERG and MSWEP may constitute such a variation in the obtained signal. Please also refer to our replies to some of the comments below.

Figure 5: Quantile--Quantile plots of the total amount of afternoon rain per event

Lines 193-198 – I see that the focus is on the downwind pattern, but I think it’s worth considering the upwind pattern more closely. For example, IMERG seems to encounter even larger changes upwind than downwind, for likelihood (Figure 2) and possibly for intensity (Figure 5). In fact, the decreases in likelihood upwind for IMERG are stronger than the increases in likelihood downwind for MSWEP (Figure 2). Therefore, it may be helpful to add more text discussing this phenomenon, since it may be on par with the downwind pattern. It may also be worth considering the *difference* between upwind and downwind likelihoods as part of this study, since the difference between downwind and upwind seems similar between MSWEP and IMERG in Figures 2 and S2 (and other figures) even though the actual values are different.

We thank the reviewer for this thoughtful comment and agree that a more in-depth discussion of the upwind signal was missing. It is apparent for both IMERG and MSWEP that rain events are less likely to be located upwind of irrigated grid cells. There is also a gradient showing that afternoon rain becomes less likely further upwind from irrigated grid cells, i.e., afternoon rain is still more likely at and near upwind irrigated areas in comparison to grid cells further away. This is particularly evident for IMERG, as previously noted, and may be related to systematic mismatches in peak rainfall locations between IMERG and MSWEP.

However, the upwind signal may also originate from (i) potential upwind moisture convergence close to irrigated areas, while (ii) some of our methodological assumptions leading to epistemic limitations could also contribute to part of the signal:

-(i) There can be conditions in which mesoscale wind patterns develop due to large-scale irrigation, leading to moisture convergence and associated convection upwind of irrigated areas. Irrigation may lead to daytime breeze circulations (Lunel et al., 2024) or mesoscale wind patterns favouring upwind moisture convergence (Lo et al., 2021). We added the following text to the Section on "Exploring Mechanisms in Irrigation-Rainfall Interactions" (l. 219-226):

Afternoon rain is still more likely over irrigated grid cells than upwind, indicating that increasing atmospheric moisture through enhanced evaporation has at least some local effect, which partly corresponds to observations of convective properties across irrigated areas within the midwestern United States (Lachenmeier et al., 2024). In addition, the detected upwind gradient (especially for IMERG) indicates that afternoon rain is becoming more likely closer to irrigated areas. One possible explanation is that irrigation alters mesoscale wind patterns, leading to moisture convergence upwind of irrigated areas.

-(iia) The majority of detected rain events include multiple irrigated grid cells within the 50km circle surrounding the peak location (see Fig. S5). That means that many rain events may also feature irrigated grid cells up and downwind of their peak location. While our results clearly show that most rain events are located downwind from irrigated grid cells, this does not exclude the fact that the same rain events can also be located at the location or upwind of an irrigated grid cell. Hence, as we detect a strong downwind signal from rain events that often feature multiple irrigated grid cells in their surroundings, we must, to a certain extent, also expect a contrasting upwind signal. We added the following text to the Section on "Exploring mechanisms" (l. 226-230):

However, most detected rain events include at least a few additional irrigated grid cells within the 50km circle surrounding the peak location (see Fig. S5). As a result, many rain events feature irrigated grid cells both upwind and downwind of their peak location. Therefore, while our analysis identifies a strong (positive) downwind signal, we must also anticipate a corresponding (negative) upwind signal to some extent..

-(iib) We consider monthly wind direction climatologies. However, not all regions are characterized by persistent wind patterns. It is possible that daily wind directions and associated up- and downwind patterns for individual rain events may vary. That means part of the gradient in the upwind signal may be an actual downwind signal under conditions of possibly shifted actual afternoon wind direction, resulting in higher afternoon rainfall likelihood closer to or at irrigated grid cells. We added the following text to the Discussion (l. 301-305):

Additionally, when considering monthly wind climatologies, daily wind directions and the associated upwind and downwind patterns for individual rain events may vary. As a consequence, part of the gradient in the upwind signal might occur due to afternoon wind direction shifts, resulting in a higher likelihood of afternoon rainfall closer to or at irrigated grid cells.

Figure 6 – I noticed in this figure that the upwind values are rather insensitive to the irrigation threshold. Does that mean that upwind areas, which have strong decreases in likelihood for IMERG, are not affected by irrigation? Or does it mean that the relationship is related to irrigation but to a different mechanism than the downwind area (like the one mentioned in lines 221-222)? There are no conclusive answers here, but I think it could be worth briefly discussing this in the manuscript.

We thank the reviewer for that interesting and intriguing observation. We agree that the upwind signal is likely driven by a different mechanism (either related to actual processes or epistemic uncertainties due to limitations in our methodology). Such a mechanism could be linked to land use changes rather than direct irrigation activities. The signal may also originate from specific features in the sample of afternoon rain events. However, we can only speculate on this or other potential

reasons as a more in-depth assessment would require high-resolution and more detailed observations of near-surface wind patterns, atmospheric moisture, boundary layer characteristics, and land surface information. However, the insensitivity of the choice of the irrigation threshold on the upwind response suggests to us that large-scale irrigation mainly affects the downwind signal. We added the following text to the Discussion (l. 286-291):

On the other hand, lowering the irrigation threshold has a minimal effect on the upwind signal, suggesting that the upwind signal may occur due to specific features in the sample of detected afternoon rain events or is influenced by different mechanisms potentially linked to land use changes rather than direct irrigation activities. In conclusion, these results indicate that large-scale irrigation activities mainly influence the downwind signal.

Lines 291-292 – I think changing from a 12-hr to 6-hr period does have a noticeable change on MSWEP in the downwind region. While the shading is greater than zero for the 2-4 grid cell range from 12-24 hr, it only seems to be greater than zero for the 4 grid cell value from 12-18 hr. So there still seems to be an effect, but it seems less robust. It would be worth mentioning this in the manuscript.

We agree with the reviewer that there is noticeable change, especially regarding MSWEP. We have changed the sentence as follows (l. 356-358):

However, please note that considering a shorter afternoon period (12h to 18h) reveals a similar signal, although it is less robust.

Lines 314-316 – Actually, the downwind signal effectively disappears when switching to the MEIER-AEI dataset. Interestingly, the IMERG upwind signal strengthens, too. Any thoughts as to why this is the case?

We want to note that the updated map of irrigated areas provided by Meier et al. (2018) globally shows 18% larger irrigated areas compared to the FAO-GMIA used in this study. This includes larger consecutive irrigated areas in many parts of South and East Asia and additional irrigated areas worldwide, especially across Central Asia (please see figure below from Meier et al., 2018). However, those additional areas were identified using NDVI data and include, as stated by the authors, many sparsely irrigated areas across South and Central Asia. We believe that additional afternoon rain events identified across the 18% additional and, in most cases, sparsely irrigated land are not influenced by actual irrigation and weaken the overall result. The updated text reads as follows (l. 377-385):

Using two other irrigation data sources providing (i) updated AEI for the year 2015 (Mehta et al., 2024) and (ii) extended data (Meier et al., 2018) showing 18% more irrigated areas worldwide in comparison to the FAO data used within this study yield a weaker signal, particularly when considering the extended data. However, it is important to note that the extended data also includes more widespread sparsely irrigated regions. Rain events identified near these areas are likely not directly influenced by large-scale irrigation activities, which contaminates the overall signal.

[REDACTED]

Irrigated areas identified based on different approaches (from Meier et al., 2018) showing additionally and more sparsely irrigated areas in South and East Asia (red colors) and Central Asia (yellow/green colors). Please see Meier et al. (2018) for a high-res version of this figure.

Lines 339-340 – How much do the detected rain events decrease with larger radii? If feasible, it could be telling to analyze whether these patterns hold with larger radii, even 60 or 70 km.

We included a Supplementary Figure illustrating results considering a 70km radius (Fig. S10). We obtain similar signals within the 50km range. Beyond 50km in the downwind range, we find a stark decline in afternoon rain likelihood, supporting our conclusion that the irrigation impact is most substantial between 30-50km downwind of irrigated areas. The upwind decline extends beyond the

Figure S10: Average likelihood of peak afternoon rain along the wind direction considering a 50km and 70km radius surrounding peak afternoon rain locations

50km range, especially for IMERG. However, it is important to note that the number of events drastically reduced by 51% (MSWEP) and 60% (IMERG). Various irrigated areas across many world regions are excluded (e.g., Central Asia, Southern Europe, East Asia, Southeast Asia). Besides motivating the 50km radius by considering average storm sizes (ca. 25km), we believe that 50km represents a good compromise between finding a meaningful signal, including a reasonably large number of events, and featuring irrigated areas across all continents.

Lines 420-423 – I'm a little confused about the methodology for choosing the "downwind" and

“upwind” areas for Figure 5. It seems that “downwind” only refers to the downwind “band” (12 grid cells) whereas almost everything else is considered “upwind”. I may be misunderstanding this, but if my understanding is correct, then the regions considered would be very different sizes. Even for Fig. S11, it seems that half of the area of interest would be considered upwind but only the downwind “band” would be considered “downwind”. Any clarification would be much appreciated.

We define "downwind" rain events as those that occur in the downwind band, meaning they are influenced by irrigation from upwind areas. In contrast, "upwind" events refer to all other rain events outside the downwind band and are not directly affected by irrigation from upwind irrigated grid cells. We agree that the previous wording may have been misleading. Therefore, we have revised the terminology throughout the manuscript. We now refer to "downwind" events as those occurring within the downwind band (as before) and have renamed the "upwind" events as "non-downwind" events. Although the regions differ in size, the number of events is roughly similar, as we detect more downwind rain events (see Fig. S15).

Figure S9 – I think it’s important to better differentiate in the figure caption how the “mean monthly wind data” in this figure differs from that of the original data (i.e., the climatology).

We revised the figure caption to clarify that we consider actual mean monthly wind data from 2001 to 2020 in Fig. S9 (Fig. S12 in the updated manuscript) while we considered the respective 20-year monthly wind climatology in all other figures.

Minor comments

Line 20 – “Yet, the effect on rainfall patterns near irrigated areas remains less clear” – compared to what?

We changed the sentence to: *Yet, the effect on rainfall patterns near irrigated areas remains unclear.*

Lines 131-133 – I understand the rationale for focusing on monthly winds, but out of curiosity, were sub-daily winds actually used within this methodology (even if it’s not ultimately shown in the manuscript)?

No, we believe that daily to sub-daily winds obtained from ERA5 at roughly 0.25deg are not suitable for the analysis carried out in this study, especially since the ERA5 land surface component does not represent irrigation activities (i.e., it may not capture sub-daily mesoscale circulation patterns surrounding irrigated areas). Additionally, to support our rationale presented in the main text, we aimed (and succeeded) at identifying a signal using more general and broad assumptions, such as monthly wind climatologies rather than daily or sub-daily wind estimates.

Lines 137-139 – This sentence is a bit confusing to me. It might be more straightforward to say that the likelihood of peak afternoon rain directly at the location of the irrigated grid cells is between the likelihoods upwind and downwind.

We rephrased the respective sentence to (l. 147-149):

This further indicates a likelihood of peak afternoon rain directly at the location of the irrigated grid cells which falls between the likelihood in downwind and upwind areas.

Line 164 – I think the figure reference “(see Fig. 4 c,d)” belongs at the end of the previous sentence, because that sentence directly refers to the results in the figure.

We provide the figure reference at the end of the previous sentence in the revised manuscript.

Figure 4 – Any thoughts as to why there is a stronger signal closer to the irrigated grid cells in the rest of the world compared to South Asia (at least for MSWEP)? Also, any thoughts as to why MSWEP is showing increased likelihood both downwind *and* upwind in South Asia?

We thank the reviewer for these very interesting questions. South Asia is mainly characterized by large, consecutive irrigated areas across the Indo-Gangetic Plain and the high mountain ranges to the north (e.g., the Himalayas). Due to these geographical features, we believe some of the weakened upwind and downwind signals we detect in South Asia can be related to our event detection methodology. As we consider a 300m elevation threshold, we basically exclude rain events taking place close to the irrigated areas along the Himalayan foothills in the northern part of the Indo-Gangetic Plain. That systematic exclusion of irrigated areas to the north of our study area, in addition to mountain valley wind patterns that may extend beyond the 50km scale considered in this study, may create the observed signals for rain events detected in South Asia. Please also refer to our response regarding the comments on regional signals made by Reviewer 1.

Lines 289-291 – Is the precipitation distribution for MSWEP similar to that of IMERG for the rain events used in this study? I’m curious whether the shorter time step for IMERG results in a different distribution.

A closer look at the table below enables a basic assessment of the precipitation distributions for MSWEP and IMERG. Both distributions have distinct differences, with MSWEP generally showing larger rain amounts except for the 99th percentile. However, we are uncertain whether these differences occur due to a shorter time step for IMERG. While MSWEP merges satellite observations with streamflow observations and other data sources for distributional bias correction, IMERG is mainly based on satellite observations. These fundamental differences may lead to systematic differences in the precipitation distributions.

Percentile	1	5	10	25	50	75	90	95	99
MSWEP	3.06	3.25	3.56	4.63	7.63	14.0	24.5	33.25	57.64
IMERG	3.03	3.15	3.31	3.87	5.34	8.72	15.8	24.3	61.02

We have added the table to Supplementary Information and the following text to Methods section (l. 343-347):

Nonetheless, we find distinct differences between the rainfall distributions of MSWEP and IMERG, with MSWEP generally showing larger rain amounts except for the 99th percentile (see Supplementary Table 1). These differences may arise from fundamental differences in the methodologies used to estimate rainfall (see above), which can lead to systematic differences in the precipitation distributions.

Lines 295-296 – Was there a particular reason for initially choosing 3 mm as the threshold value?

We refer to the 3 mm threshold used by Taylor et al. (2012) in their study assessing the increased likelihood of afternoon rain over drier soils. We now cite this study to justify our choice.

Lines 303-305 – One thing to consider: Even if the total irrigated grid cell percentage within 50-km of a rain event is <20%, it's possible that many grid cells upwind of the center location are also irrigated. Perhaps that's related to the strong decreases in likelihood in IMERG.

We thank the reviewer for this interesting suggestion, which we already considered in our response to a previous comment. We added the following text (l. 226-230):

However, most detected rain events include at least a few additional irrigated grid cells within the 50km circle surrounding the peak location (see Fig. S5). As a result, many rain events feature irrigated grid cells both upwind and downwind of their peak location. Therefore, while our analysis identifies a strong (positive) downwind signal, we must also anticipate a corresponding (negative) upwind signal to some extent.

Lines 401-403 – The aridity dataset seems to only cover 1970-2000, but the MSWEP and IMERG datasets cover 2001-2020. Is it possible that aridity experienced major changes since 2000 that would majorly affect these results?

We use data from Version 3 of the Global Aridity Index and Potential Evapotranspiration Database (Zomer et al., 2022) as it provides a state-of-the-art high-resolution global aridity index estimate. The periods do not overlap (1970–2000 for the aridity index and 2001–2020 in our study). However, we intended to use a robust and widely used climatological aridity index estimate that represents aridity conditions at the beginning of our study period. We agree with the reviewer that there might have been changes in aridity since 2000. While we do not believe that these changes substantially alter our assessment, accounting for these changes would require a separate evaluation of aridity estimates, which goes beyond the scope of our study. We added the following to the Methods section to clarify that we use climatological aridity estimates from a time preceding our study period (l. 477-481):

We use AI estimates from the Global ETO and Aridity Index Database Version (Zomer et al., 2022) providing high-resolution (30") geospatial data of averaged aridity conditions for the 1970–2000 period. While this dataset precedes our study period (2001–2020), AI is a climatological feature representing historical conditions of dryness and wetness and is used here for classification purposes.

Line 404 – Should “targeted” be “target”?

Changed.

Lines 409-411 – Since it seems that the two current aridity categories overlap for “semi-arid”, I wonder if it would be helpful to conduct the analysis with three aridity categories that more cleanly cover the different aridity conditions. If such an analysis provides useful results, the results could be substituted for Figure 4 or provided as a new supplementary figure.

We have included a new Supplementary Figure (Fig. S14) showing a separate analysis for arid ($AI < 0.2$), semi-arid ($0.2 < AI < 0.5$), and humid ($AI > 0.5$) regions. We detect the strongest downwind

signal under arid conditions, whereas the downwind signal becomes much weaker under humid conditions. However, we have decided to keep Fig. 4 as it is, mainly because the sample sizes differ substantially between the individual aridity categories. Dividing the data into equally sized subsamples (as in Fig. 4) allows for a more effective comparison.

Figure S14: Average likelihood of peak afternoon rain along the wind direction for rain events detected in (a) arid ($AI < 0.2$), (b) semi-arid ($0.2 < AI < 0.5$), and (c) humid ($AI > 0.5$) regions.

We added the following text to the Methods section (l. 489-492):

Please refer to Fig. S14 for the respective results in arid ($AI < 0.2$), semi-arid ($0.2 < AI < 0.5$), and humid ($AI > 0.5$) regions. We detect the strongest downwind signal under arid conditions, whereas the downwind signal becomes much weaker under humid conditions.

Lines 445-447 – Regarding the code: The title on the Zenodo page doesn't match the title of the manuscript – I'm not sure if that's intentional. Also, the files are Restricted, so I can't actually view the code files. I don't think seeing the code is essential for this review, but I hope that the code will be made available by the time of publication.

Yes, the different titles were intentional. The title on Zenodo more accurately reflects the functionality of the associated scripts and notebooks. The study is mentioned in the data description and the README file, and we will add the corresponding link once the study is accepted.

Line 533 – There seems to be a typo on this line.

Changed.

Figure S10 – The "x" in the center of panel b) is almost invisible. It would be probably helpful to change its color (and perhaps that of panel a)) to improve the contrast.

We agree that the "x" in the center was almost invisible. We changed the color accordingly.

Reviewer #1 (Remarks to the Author):

The authors have addressed most of my comments. I have one more comment for the authors to consider. As can be seen in Figs. 3 and 4, the mean peak location of afternoon rain relative to irrigated grid cells is at around 3 grid cell distance. Thus, my question is what determines the distance and how the distance changes in different regions/occasions. Possible factors include wind speed, surface aridity, and topography. Hope the authors can provide some insight or analysis on this.

We sincerely thank the reviewer for their positive evaluation of the revised manuscript. We are pleased to have addressed all previous comments successfully. Regarding the peak in afternoon rainfall likelihood observed approximately 3-4 grid cells downwind of irrigated areas, we agree that this is an intriguing result requiring further discussion.

While the peak is evident in MSWEP data, we also observe a peak using IMERG data at around 50 km downwind of irrigated grid cells when considering a larger radius of 70 km. From a regional perspective (Fig. S4), this peak is detectable in North America (MSWEP), South/Central Europe (MSWEP), East Asia (IMERG), and South Australia (MSWEP). In other regions, the peak may occur outside the 50 km radius or may not be evident due to large uncertainties and small sample sizes.

Our interpretation is that the robust detection of a peak in afternoon rainfall likelihood between 30 to 50 km downwind indicates a clear local irrigation-atmosphere feedback impacting the occurrence of afternoon rain. However, varying land surface and atmospheric characteristics may impact the peak location. For example, the peak is evident in semi-arid environments, whereas under arid conditions, the peak may be located outside the considered 50 km radius (Fig. S14).

We can only speculate on potential reasons for these differences, but one might conclude that the footprint of irrigation-afternoon rainfall interactions might be larger under arid conditions and more variable under more humid conditions. Additionally, the peak location does not seem to be affected by the irrigation threshold (Fig. 5) or any other thresholds (Fig. S7).

We added the following text to the Discussion (l. 276-282):

We detect a peak in afternoon rainfall likelihood approximately 3 grid cells downwind of irrigated areas for MSWEP and about 4 grid cells downwind for IMERG (see Fig. S6). This indicates a clear local irrigation-atmosphere feedback that affects the occurrence of afternoon rain within 50 km of irrigated areas. However, varying land surface and atmospheric characteristics may alter the peak location. For example, while the peak is evident in some regions (e.g., North America, East Asia), it may be found outside the considered 50 km radius in other areas (see Fig. S4).

Reviewer #2 (Remarks to the Author):

I very much appreciate the authors comprehensively addressing all of the comments from my previous review. I believe that this manuscript is nearly ready for acceptance into Nature Communications. Following are some minor additional comments:

We sincerely thank the reviewer for their positive evaluation of the revised manuscript. We are pleased to have successfully addressed all previous comments. Please see our detailed responses to the remaining minor comments below.

Line 190 - I think it would be helpful to be consistent with the terminology regarding the 12-cell downwind "band" that is used for Figure 5 (and Figure 3c,d), especially since this term is already used in the Methods section (e.g., line 451). I would change "section" to "band" in this line and add the term "upwind and downwind bands" somewhere within this parenthetical expression in the caption of Figure 3: "(+/-1 grid cells, within the dashed horizontal lines in a, b)".

Thank you for your suggestion. We now refer to the "downwind band" consistently throughout the manuscript and have updated the figure caption for Fig. 3 accordingly.

Figure 5 - Please change "Upwind Rain" to "Non-Downwind Rain" for the y-axis label and change "downwind section" to "downwind band" in the figure caption.

We have revised the figure and updated the caption accordingly.

Lines 294-295 - I see that the word "upwind" was added here, but I'm a little confused because the "signal is less pronounced" only for the downwind band. Any clarification would be much appreciated.

We agree with the reviewer that the term "upwind" can cause confusion, so we have removed it. We now conclude that the signal itself is associated with afternoon rain, not just the downwind signal.

Lines 401-402 - Actually, the signal seems to go slightly negative (not just weaker) for humid conditions. I would adjust this sentence to reflect this.

We assume the reviewer is referring to lines 491-492 here. The revised sentence now reads like: *We detect the strongest downwind signal under arid conditions, whereas the downwind signal becomes slightly negative under humid conditions.*

Figure S2 - Optional, but it might be helpful to change the color at the upper boundary of the colorbar to something that looks a little more different from the color of the water bodies in the map.

We thank the reviewer for this suggestion. To prevent confusion, we have decided to adjust the color of the water bodies to a slightly more bluish tone in Figures 2, S1, and S2.

Figure S10 - I'm a little confused by the figure caption, because while it indicates that the 70-km data is dark green, it actually seems that the original (50-km) data is dark green. The 70-km data seems to be more of a dark gray color. It would be helpful to clarify this color in the caption.

Thank you for bringing the color confusion to our attention. We recognize that the very dark green we used for the SI figures appears dark gray due to the added transparency for the lines and shading. We have updated the figure captions in the SI accordingly.